# The *Aspergillus fumigatus* transcription factor RglT is important for gliotoxin biosynthesis and self-protection, and virulence

Laure N. A. Ries[1], Lakhansing Pardeshi[2,3], Zhiqiang Dong[3], Kaeling Tan[2,3,4], Jacob L. Steenwyk[5], Ana Cristina Colabardini[6], Jaire A. Ferreira Filho[6], Patricia A. de Castro[6], Lilian P. Silva[6], Nycolas W. Preite[7], Fausto Almeida[1], Leandro J. de Assis[6], Renato A. C. dos Santos[6], Paul Bowyer[8], Michael Bromley[8], Rebecca A. Owens[9], Sean Doyle[9], Marilene Demasi[10], Diego C. R. Hernández[6], Luís Eduardo S. Netto[11], Monica T. Pupo[6], Antonis Rokas[5], Flavio V. Loures[7], Koon H. Wong[2,12]*, Gustavo H. Goldman[6]*

1 Faculty of Medicine of Ribeirao Preto, University of São Paulo, Ribeirão Preto, Brazil, 2 Genomics and Bioinformatics Core, Faculty of Health Sciences, University of Macau, Macau SAR, China, 3 Faculty of Health Sciences, University of Macau, Macau SAR, China, 4 Centre for Precision Medicine and Research and Training, University of Macau, Macau SAR, China, 5 Department of Biological Sciences, Vanderbilt University, Nashville, TN, United States of America, 6 Faculty of Pharmaceutical Sciences of Ribeirao Preto, University of São Paulo, Ribeirão Preto, Brazil, 7 Institute of Science and Technology, Federal University of São Paulo, São José dos Campos, Brazil, 8 Manchester Fungal Infection Group, Division of Infection, Immunity and Respiratory Medicine, School of Biological Sciences, Faculty of Biology, Medicine and Health, University of Manchester, Manchester, United Kingdom, 9 Department of Biology, Maynooth University, Maynooth, Ireland, 10 Institute Butantan, Laboratory of Biochemistry and Biophysics, São Paulo, Brazil, 11 Institute Biosciences, University of São Paulo, São Paulo, Brazil, 12 Institute of Translational Medicine, University of Macau, Macau SAR, China

* koonhowong3@gmail.com (KHW); ggoldman@usp.br (GHG)

**Data Availability Statement:** Short reads were submitted to the NCBI's Sequence Read Archive under accession number SRP154617 (https://

## Abstract

*Aspergillus fumigatus* is an opportunistic fungal pathogen that secretes an array of immune-modulatory molecules, including secondary metabolites (SMs), which contribute to enhancing fungal fitness and growth within the mammalian host. Gliotoxin (GT) is a SM that interferes with the function and recruitment of innate immune cells, which are essential for eliminating *A. fumigatus* during invasive infections. We identified a C6 Zn cluster-type transcription factor (TF), subsequently named RglT, important for *A. fumigatus* oxidative stress resistance, GT biosynthesis and self-protection. RglT regulates the expression of several *gli* genes of the GT biosynthetic gene cluster, including the oxidoreductase-encoding gene *gliT*, by directly binding to their respective promoter regions. Subsequently, RglT was shown to be important for virulence in a chemotherapeutic murine model of invasive pulmonary aspergillosis (IPA). Homologues of RglT and GliT are present in eurotiomycete and sordariomycete fungi, including the non-GT-producing fungus *A. nidulans*, where a conservation of function was described. Phylogenetically informed model testing led to an evolutionary scenario in which the GliT-based resistance mechanism is ancestral and RglT-mediated regulation of GliT occurred subsequently. In conclusion, this work describes the function of a previously uncharacterised TF in oxidative stress resistance, GT biosynthesis and self-protection in both GT-producing and non-producing *Aspergillus* species.

www.ncbi.nlm.nih.gov/sra/?term=SRP154617).
The ChIPseq data are available from NCBI SRA
(sequence read archive) database under accession
number PRJNA574873 (https://www.ncbi.nlm.nih.
gov/Traces/study/?acc=PRJNA574873&o=acc_s%
3Aa).

**Funding:** Fundação de Amparo à Pesquisa do
Estado de São Paulo (FAPESP), grant numbers
2017/14159-2 (LNAR), 2016/07870-9 (GHG),
2017/01188-4 (DCRH), 2013/50954-0 (MTP),
2018/25217-6 (JAFF), 2014/00789-6 (LJA), 2018-
14762-3 (FVL), 2017/21983-3 (RACS), 2013/
07937-8 (Redox Processes in Biomedicine, MD,
LESN), 2019/09278-8 (NWP), 2016/03322-7 (FA),
2017/07536-4 (ACC) and the Conselho Nacional de
Desenvolvimento Científico e Tecnológico (CNPq)
(301058/2019-9 and 404735/2018-5 to GHG and
303792/2018-2 to MTP). LNBR - Brazilian
Biorenewables National Laboratory (CNPEM/
MCTIC) during the use of the High Throughput
Sequencing (NGS) open access facility in
generating the RNA sequencing data described
here. Coordenação de Aperfeiçoamento de Pessoal
de Nível Superior – Brasil (CAPES) – Finance Code
001. Wellcome Trust grant 208396/Z/17/Z (PB,
MB). High Performance Computing Cluster (HPCC)
supported by Information and Communication
Technology Office (ICTO) of the University of
Macau. KHW is supported by the Research
Services and Knowledge Transfer Office (RSKTO)
of the University of Macau (grant number:
MYRG2018-00017-FHS), Institute of Translational
Medicine and Faculty of Health Sciences internal
funds. LP, ZD and KT are supported by the Centre
for Precision Medicine Research and Training and
Faculty of Health Sciences of the University of
Macau. The authors also received support from a
Collaborative Research Fund Equipment Grant
(C5012-15E) from the Research Grant Council,
Hong Kong Government. This work was also
supported in part by grants from the National
Science Foundation (DEB-1442113 to AR),
Vanderbilt University (Discovery grant to AR), and
by the Howard Hughes Medical Institute through
the James H. Gilliam Fellowships for Advanced
Study program (JLS and AR). This work used
resources of the "Centro Nacional de
Processamento de Alto Desempenho em São
Paulo (CENAPAD-SP). The funders had no role in
study design, data collection and analysis, decision
to publish, or preparation of the manuscript.

**Competing interests:** The authors have declared
that no competing interests exist.

## Author summary

*A. fumigatus* is a prevalent opportunistic fungal pathogen that causes a wide range of diseases, in both immunocompetent and immunocompromised subjects. Contributing to *A. fumigatus* virulence, is the *in vivo* secretion of gliotoxin, a mycotoxin with both immuno-suppressive and immunomodulatory properties. Regulation of the gliotoxin-encoding biosynthetic gene cluster has proven to be a multifactorial and hierarchical process. In this work, we identify a transcription factor, named RglT (regulator of gliotoxin), that is crucial for gliotoxin biosynthesis and self-protection, a process that mainly occurs through the direct regulation of the oxidoreductase-encoding gene *gliT*. The absence of *rglT* resulted in a strain that was highly sensitive to oxidative stress, to exogenous gliotoxin and hypovirulent in a chemotherapeutic murine model of IPA. Furthermore, we show that homologues of both RglT and GliT co-exist within the class of eurotiomycetes and appear to be functionally conserved in non gliotoxin-producing fungal species. Our study therefore highlights the importance of mycotoxin self-protection in pathogenic and non-pathogenic fungal species with consequences for fungal virulence and survival.

## Introduction

*Aspergillus fumigatus* is an opportunistic fungal pathogen for invasive disease in immunocompromised individuals, a secondary pathogen in immune competent individuals (e.g. chronic pulmonary aspergillosis) and probably the primary causative agent of allergic disease in atopic individuals[1]. Resident alveolar macrophages of the innate immune system are the first line of defence against invading *A. fumigatus* conidia[2]. Upon phagocytosis of conidia, macrophages release an array of chemokines and cytokines, resulting in the recruitment of additional white blood cells, such as neutrophil granulocytes[2,3]. Neutrophil granulocytes are the most abundant circulating immune cells and have been shown to be crucial for the immune response against *A. fumigatus* [2,3]. Neutrophils can rapidly kill *A. fumigatus* germinating conidia via both non-oxidative and oxidative mechanisms [3,4]. Non-oxidative mechanisms include degranulation, whereby different types of granules, containing enzymes with fungicidal activities and iron-chelating compounds, are released [3,4]. The respiratory burst exerted by neutrophils results in increased oxidative stress through generation of reactive oxygen species (ROS), a process that depends on the assembly and activity of the multi-subunit NADPH oxidase, and in the production of hypochlorous acid by the primary granule-resident enzyme myeloperoxidase [3,4]. Furthermore, neutrophils also form neutrophil extracellular traps (NETs), extracellular structures that consist of DNA decorated with antimicrobial proteins, thought to be mainly active against large fungal structures such as hyphae, but which have been suggested to not promote fungal killing but rather prevent pathogen spreading *in vivo*[5].

To resist and protect themselves from the fungicidal oxidative activities of the host immune system, *A. fumigatus* employs several counter-measures including the secretion and production of an array of immune-modulatory factors, such as secondary metabolites (SMs), that contribute to enhancing fungal fitness and growth during infection[6]. The SM gliotoxin (GT) is well-characterised in *A. fumigatus* and was shown to interfere with macrophage-mediated phagocytosis through prevention of integrin activation and actin dynamics, resulting in macrophage membrane retraction and failure to phagocytose pathogen targets[7]. Furthermore, GT inhibits the production of pro-inflammatory cytokines secreted by macrophages and the activation of the NFkB regulatory complex[3]. GT also interferes with the correct assembly of

NADPH oxidase through preventing p47phox phosphorylation and cytoskeletal incorporation as well as membrane translocation of subunits p47phox, p67phox and p40phox[8]. Recently, GT was also identified as an inhibitor of neutrophil chemoattraction by targeting the activity of leukotriene A4 (LTA$_4$) hydrolase, an enzyme that participates in LTA biosynthesis[9]. The immune-suppressive and–modulatory properties of GT thus mainly affect innate macrophage and neutrophil responses[3] and the presence of this SM has been detected *in vivo* in both murine models of invasive aspergillosis (IA) and in human cancer patients with documented IA[10].

The biosynthesis and secretion of GT are catalysed by an array of enzymes that are encoded by a gene cluster comprised of 13 *gli* genes on chromosome VI[11]. A sequence of enzymatic steps results in the formation of dithiol gliotoxin with the oxidoreductase GliT catalysing disulphide bridge closure, resulting in the production of the final toxic form of GT[11,12]. Regulation of the *gli* genes is governed by a myriad of different proteins, such as transcription factors (TFs), transcriptional regulators, chromatin-associated factors, the MAP kinase MpkA, developmental regulators and regulators of G-protein signalling[11]. The GT cluster resident TF GliZ also controls the expression of some *gli* genes, excluding *gliT*, which is essential for self-protection against exogenous GT[12]. GT biosynthesis regulation is therefore a hierarchical and multifactorial process.

In this work, we identify a TF (Afu1g09190) that is essential for *A. fumigatus* GT biosynthetic gene regulation and virulence. Deletion of this TF, named RglT (Regulator of gliotoxin), conferred oxidative stress sensitivity and abrogated resistance against exogenously-added GT and of GT biosynthesis, mainly through the direct regulation of *gliT*. Furthermore, RglT was shown to be important for virulence in chemotherapeutic mice with IPA. Homologues of this TF are present in fungi belonging to the classes of Eurotiomycetes and Sordariomycetes, including the non-GT-producing fungus *A. nidulans*, where it is important for oxidative stress resistance and essential for GT self-protection through the regulation of a gene encoding a putative GliT homologue. Furthermore, we predict that GliT-resistance mechanism evolved first and RglT-mediated regulation of *gliT* occurred thereafter.

## Results

### Identification of a transcription factor that confers resistance to oxidative stress

Initially, this work aimed at uncovering previously uncharacterised TFs that are important for carbon catabolite repression (CCR), through screening an *A. fumigatus* transcription factor deletion library for growth defects in the presence of allyl alcohol (AA), a compound indicative for potential defects in CCR[13,14]. A strain that was deleted for gene Afu1g09190 (AFUB_008610) was highly sensitive to AA (Fig 1A and 1B) but presented no growth defects in non-stress conditions (S1A and S1B Fig) nor in the presence of 2-deoxyglucose (2DG), an additional indicator for defects in CCR[15] (S1C Fig). AA is converted by alcohol dehydrogenase (ADH) to acrolein, a highly toxic compound shown to also induce oxidative stress within the cell[16,17]. To investigate any ADH regulatory or activity defects, enzyme activity was measured in the presence of ethanol and ethanol and glucose, with glucose causing the CCR-mediated transcriptional repression of ADH-encoding genes[18]. ADH was not significantly different between the parental-type (referred to as wild-type, WT) CEA17 strain and the ΔAfu1g09190 strain in all conditions (S1D Fig), suggesting that the observed sensitivity of strain ΔAfu1g09190 was not due to regulatory defects in ADH activity and in CCR.

We therefore decided to determine whether strain ΔAfu1g09190 was also sensitive to other oxidative stress-inducing compounds. Growth was characterised in the presence of acrolein,

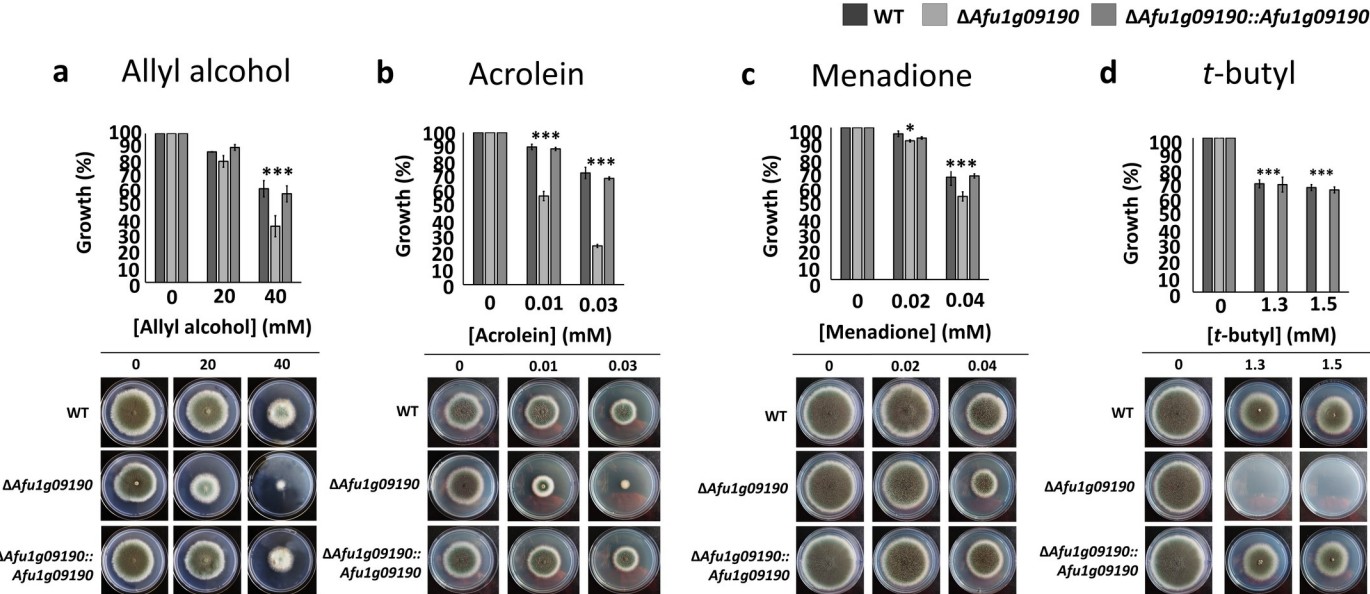

**Fig 1. The transcription factor encoded by Afu1g09190 is important for resistance against oxidative stress. a-d**, Strains were grown for 5 days from $10^5$ spores at 37°C on glucose minimal media supplemented with increasing concentrations of the oxidative stress-inducing compounds (**a**) ally alcohol, (**b**) acrolein, (**c**) menadione and (**d**) *t*-butyl hydroperoxide (*t*-butyl). Graphs indicate the % of growth in the presence of the respective drug with respect to the control condition (without drug). Graphs are the quantitation of radial growth of the pictures shown in the same panel, with standard deviations representing three biological replicates (\*P-value < 0.01; \*\*P-value < 0.001; \*\*\*P-value < 0.0001 in a two-way ANOVA test).

menadione and *t*-butyl hydroperoxide (*t*-butyl). Growth of strain ΔAfu1g09190 was significantly inhibited in the presence of all the aforementioned compounds, when compared to the WT and complemented strains (Fig 1C–1F). The aforementioned results therefore suggest that the *A. fumigatus* TF Afu1g09190 is important for mediating resistance against different types of oxidative stresses.

## Gliotoxin (GT) biosynthesis-related genes are under the transcriptional control of Afu1g09190

To gain understanding of the cellular processes governed by Afu1g09190, RNA-sequencing was carried out for the wild-type (WT) and ΔAfu1g09190 strains, when grown for 24 h in GMM (glucose minimal medium, control) and after the addition of 10 mM AA for 30 min. The Afu1g09190 TF was confirmed to localise to the nucleus in the presence of AA (S1E Fig) by microscopy analysis of a GFP-tagged Afu1g09190 strain (S1F and S1G Fig).

In the first series of analyses, the transcriptional response to AA in the WT strain was determined. A total of 4,563 (2,365 up-regulated, 2,198 down-regulated, log2FC ≥ 1.0 and ≤ -1.0, P-value < 0.05) genes were significantly differentially regulated in the WT strain between the control and AA condition (S1 File). Functional categorization (FunCat) (https://elbe.hki-jena.de/fungifun/fungifun.php) analyses (P-value < 0.05) of all significantly up-regulated genes showed enrichment for cellular processes such as oxidative stress response, oxygen and radical detoxification, protein folding and stabilization, detoxification, stress response and glutathione conjugation reaction (Fig 2A, S1 File). Glutathione (GSH) is a cellular antioxidant that provides protection against intra- and extracellular oxidative stresses[19]. To determine whether the observed sensitivity of the ΔAfu1g09190 strain to AA was due to a defect in GSH metabolism, intracellular GSH levels were quantified in both the WT and mutant strains when exposed to AA for different amounts of time. No difference in intracellular GSH

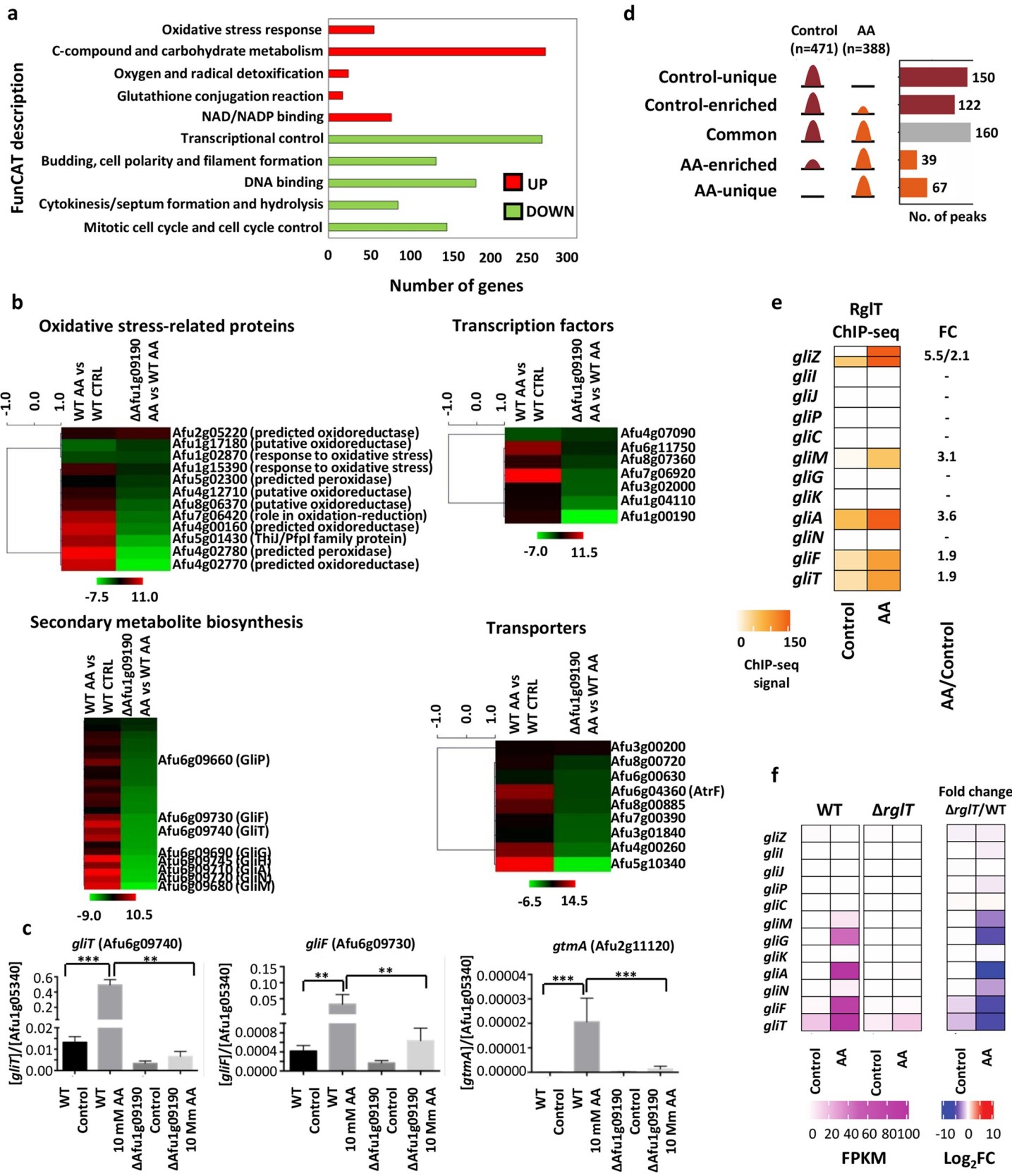

**Fig 2. The Afu1g09190-encoded TF regulates the transcription of gliotoxin (GT) biosynthetic genes and of *gtmA* through directly binding to the promoter regions in the presence of allyl alcohol (AA). a**, AA induces transcriptional up-regulation of genes required for the oxidative stress response whilst down-regulating genes encoding proteins involved in growth in the *A. fumigatus* wild-type (WT) strain. Functional categorisation (FunCat) description of the 5 most significant enriched categories (*p*-value < 0.0005), that contain up- (UP) or down (DOWN)-regulated genes with a -1 > Log2 fold-change > 1 (*p*-value < 0.05), as identified by RNA-sequencing. The WT strain was exposed to 10 mM AA for 30 min after 24 h growth in glucose minimal medium (GMM, control) and gene expression was compared between both conditions. **b**, Fumagillin and GT biosynthetic genes are severely repressed in the ΔAfu1g09190 strain. Heatmap of the Log2FC values, as determined by RNA-sequencing, of genes that were significantly (*p*-value < 0.05) differently (-1 > Log2FC < 1) expressed between the WT and Δ*rglT* in the presence of 10 mM AA for 30 min. Genes were classified into virulence-relevant categories, such as oxidative stress-related proteins, transcription factors, secondary metabolite biosynthesis and transporters. Log2FC gene expression values are also shown for the WT strain, when comparing gene expression in the presence of AA to the control condition. Hierarchical clustering was performed in MeV (http://mev.tm4.org/), using Pearson correlation with complete linkage clustering. **c**, Validation of the RNA-sequencing data by RT-qPCR. RNA was extracted from strains grown for 24 h in GMM (control) before 10 mM AA was added for 30 min and RT-qPCR was performed. Gene expression values were normalized by gene Afu1g05340, whose expression remained constant across all conditions, as confirmed by RNA-seq. Standard deviations represent three biological replicates (\*\*P-value < 0.001; \*\*\*P-value < 0.0001 in a one-way ANOVA test). **d**, Chromatin immunoprecipitation coupled to DNA sequencing (ChIP-seq) of the WT and Afu1g09190::HA strains, when grown for 24 h in GMM (control) and then exposed to 10 mM AA for 30 min, identified 538 RglT::HA binding peaks that could be classified into three main groups: i) unique (150) or enriched binding (122) in the control condition, ii) consistent binding (160) in both the control and AA conditions and iii) unique (67) or enriched (39) binding in the AA condition. **e**, Afu1g09190 binds to the promoter regions of *gliZ, gliM, gliA, gliF* and *gliT* and was subsequently named RglT (regulator of GT). Heat map depicting ChIP-seq signal for all genes of the GT biosynthetic gene cluster in the control and AA conditions. Also shown is the differential binding analysis fold change (FC) between both conditions. **f**, RNA-seq results are in agreement with the ChIP-seq results, confirming that GT biosynthetic genes are under the direct transcriptional control of RglT. Heat maps depicting RNA-seq FPKM values for the WT and Δ*rglT* strains as well as Log2FC (fold change) values between the WT to the Δ*rglT* strain of all *gli* genes in the control and AA conditions.

concentrations was observed between the WT, ΔAfu1g09190 and Afu1g09190 complemented strains in the presence of AA (S1H Fig). FunCat analysis (p-value < 0.05) of all down-regulated genes showed enrichment mainly for DNA metabolism (transcription, repair) and growth (budding, filamentation, cell cycle) (Fig 2A, S1 File). These results indicate that the addition of AA caused a fungal transcriptional response that aims at dealing with increased oxidative stress whilst down-regulating growth-related processes.

Subsequently, gene expression comparison between the WT and the ΔAfu1g09190 strains was carried out in the control and AA conditions. In the control condition, a total of 338 genes (166 genes up-regulated, 172 genes down-regulated) were significantly differentially regulated, whereas exposure to AA resulted in 115 genes (17 genes up-regulated, 98 genes down-regulated) as significantly differentially regulated between both strains (S1 File). We then further focused on genes that were significantly differentially regulated in the mutant strain, when compared to the WT strain, in the presence of AA and manually classified them into functional categories that have been reported to be important for *A. fumigatus* virulence, including cell signalling components, oxidative stress-related proteins, secondary metabolite biosynthesis, transcription factors and transporters (S2 File and Fig 2B). Of particular interest was the significant down-regulation of several genes, including the genes required for self-protection from each SM, of the fumagillin and gliotoxin (GT) biosynthetic gene clusters in the ΔAfu1g09190 strain in the presence of AA (S2 File). Subsequently, we determined the MIC (minimal inhibitory concentration) of fumagillin and GT on the WT and ΔAfu1g09190 strains. There was no difference in the MIC of fumagillin for the WT and ΔAfu1g09190 strains (100 μg/ml), whereas the ΔAfu1g09190 strain (35 μg/ml) was significantly (*p*-value < 0.05 in a 1-way ANOVA test) more sensitive to GT when compared to the WT (70 μg/ml) strain (Table 1). We therefore decided to focus on characterising the role of Afu1g09190 in GT

**Table 1. Minimal inhibitory concentrations (MIC) of fumagillin and gliotoxin on the growth of the wild-type (WT) and ΔAfu1g09190 strains.** Standard deviations are shown for 3 independent replicates for the two strains in the presence of each mycotoxin.

| Strain | Fumagillin (μg/ml) | Gliotoxin (μg/ml) |
|---|---|---|
| WT | 100 ± 0 | 70 ± 0 |
| ΔAfu1g09190 | 100 ± 0 | 35 ± 0 |

biosynthetic regulation. We identified 8 (*gliP* 3.5 log2-fold reduction; *gliF* -5.4 log2-fold reduction; *gliT* -5.5 log2-fold reduction, *gliG* -6.6 log2-fold reduction, *gliH* 6.7 log2-fold reduction, *gliA* 7.0 log2-fold reduction, *gliN* 7.2 log2-fold reduction and *gliM* 8.5 log2-fold reduction) out of the 13 GT-related biosynthesis genes that were severely reduced in expression in the ΔAfu1g09190 strain in the presence of AA (S2 File, Fig 2B). Furthermore, the expression of *gtmA* (Afu2g11120, 5.6-fold decrease), that codes for a bis-thiomethyltransferase involved in the down-regulation of GT biosynthesis through catalysing the formation of bisdethiobis (methylthio)gliotoxin (BmGT) from dithiogliotoxin[20], was also significantly reduced in the mutant strain (S2 File, Fig 2B). The RNA-seq data was validated by RT-qPCR of three randomly chosen genes (Fig 2C). These results suggest that the transcription factor encoded by Afu1g09190 is important for the expression of GT biosynthesis-related genes in the presence of AA.

## Afu1g09190 binds to GT biosynthetic gene promoter regions

To discover whether GT biosynthetic gene promoter regions are directly contacted by Afu1g09190, genome-wide binding sites were determined by ChIP-seq (chromatin immunoprecipitation coupled to DNA sequencing) of the WT (non-tagged control) and the Afu1g09190:3xHA (S1F and S1G Fig) strains in the same conditions as described for the RNA-seq analysis. Distinct binding peaks were observed throughout the genome for the Afu1g09190:3xHA strain, but not for the untagged strain, in both the control and AA conditions (S2A Fig). A total of 538 sites were bound by Afu1g09190 genome-wide under the two conditions. Among these, 160 sites were constitutively bound by Afu1g09190::3xHA (*i.e.* common targets) with the remaining 272 and 106 binding events being unique to or enriched in the control and AA conditions, respectively (Fig 2D; S3 File). Systematic assignment of intragenic and intergenic binding sites to nearest genes revealed that the Afu1g09190 intragenic binding sites were primarily located at the 5' UTR (untranslated) region of the genes (219 sites) as opposed to very few binding sites at 3' UTR (35 sites) regions and within exons (18). On the other hand, 130 intergenic binding sites were within promoter regions [-500 bp to TSS (transcription start site)] and the remaining 139 intergenic sites were further upstream than 500 bp from the TSS. Fifteen Afu1g09190 binding sites were not associated with any gene as they were at convergent gene intergenic regions. Altogether, 481 unique genes were annotated with Afu1g09190 peaks.

Next, GO (gene ontology) analysis was performed on the target genes. Gene targets that were unique to or enriched in the control condition encode proteins with functions in pathogenesis, various metabolic pathways and asexual development (S2B Fig and S3 File). On the other hand, genes with unique or increased Afu1g09190:3xHA binding in the presence of AA are significantly enriched with functions in biosynthetic pathways for GT, mycotoxin, cysteine, isoleucine and beta-glucan; as well as in oxidative stress response and oxidant detoxification (S2B Fig). These results are consistent with the above findings that Afu1g09190 is important for oxidative stress resistance and regulation of genes required for GT biosynthesis.

Indeed, differential binding analysis (DiffBind) revealed that Afu1g09190 bound to the promoter regions of the transcription factor-encoding gene *gliZ* (2.14 log2-fold stronger binding in the AA condition when compared to the control condition), the O-methyltransferase-encoding gene *gliM* (3.11 log2-fold stronger binding in the AA condition when compared to the control condition), the efflux transporter-encoding gene *gliA* (3.62 log2-fold stronger binding in the AA condition when compared to the control condition), the bidirectional promoter region of *gliT* and *gliF* (1.87 log2-fold stronger binding in the presence of AA than when compared to the control condition) (Fig 2E, S3 File). Afu1g09190 also bound to the promoter

region of *gtmA* (1.75 log2-fold stronger binding in the AA condition when compared to the control condition) in the presence of AA (S3 File). This increase in gene promoter binding correlated with a strong induction in gene expression in the WT strain in the presence of AA as shown by RNA-seq; but this induction is abolished in the ΔAfu1g09190 strain (Fig 2F). We therefore named this transcription factor RglT (Regulator of gliotoxin). These results suggest that RglT-mediated regulation of the expression of the GT biosynthetic genes *gliP*, *gliG*, *gliH* and *gliN* is indirect, whereas RglT modulates the expression of the GT biosynthetic genes *gliF*, *gliT*, *gliM* and *gliA* as well as of *gtmA* in the presence of AA through directly binding to the respective gene promoter regions.

## RglT is crucial for GT biosynthesis

Due to the observed transcriptional down-regulation of several *gli* genes, we wondered whether the Δ*rglT* strain was able to produce GT, through performing HPLC on culture supernatants of the WT, Δ*rglT* and Δ*rglT::rglT* strains, when grown in GT-producing conditions. Strains were grown in Czapek-Dox broth for 72 h as this medium has previously been shown to result in detectable gliotoxin levels in culture supernatants[12]. The Δ*gliT* strain, which does not produce GT[12], was used as a negative control. GT eluted at a retention time of 8.45 min and chromatograms showed a peak corresponding to GT in all strains except for the Δ*gliT* strain (Fig 3A). In contrast, the UV-vis absorption spectra for GT in the supernatants of the Δ*rglT* and Δ*gliT* strains were different from the standard, WT and *rglT* complemented strains (Fig 3B), indicating that the Δ*rglT* strain did not produce GT but rather a metabolite that had a similar retention time as GT. To determine the nature of this metabolite, high-resolution LC-micrOTOF-MS analysis was carried out on the same samples and the presence of a molecular ion with *m/z* 357.0945 ($[M+H]^+$) was detected in all samples, including the Δ*gliT* strain (S3A Fig). The accurate mass values of this metabolite correspond to a predicted molecular formula of $C_{15}H_{21}N_2O_4S_2$, indicating the biosynthesis of bisdethiobis(methylthio)-gliotoxin (BmGT) by Δ*rglT*[21]. BmGT was also detected in the WT, *rglT* complemented and Δ*gliT* strains (S3A Fig). LC-micrOTOF-MS detected GT in the supernatants of the WT and *rglT* complemented strains but not in the supernatants of the Δ*rglT* and Δ*gliT* strains (S3B Fig, Fig 3C and 3D). These results indicate that the transcription factor RglT is essential for GT biosynthesis in *A. fumigatus*.

## RglT directly regulates GT biosynthetic genes in GT-inducing conditions

To determine whether GT biosynthesis-related gene expression and RglT binding dynamics follow a similar pattern in GT-inducing conditions as observed for AA oxidative stress-inducing conditions, qRT-PCR and ChIP-qPCR analyses were carried out in these conditions. The expression of *gliZ*, *gliT*, *gliF* and *gtmA*, was severely reduced in the Δ*rglT* strain when grown in GT-inducing conditions (Fig 3E). We included the GT biosynthesis cluster-resident transcription factor GliZ[22], which is essential for GT biosynthesis, in our analysis in order to describe a potential transcriptional dependency between *gliZ* and *rglT* in GT-producing conditions, as binding of RglT to the *gliZ* promoter region was already observed in the presence of AA. Subsequently, we carried out ChIP-qPCR to determine whether RglT binds to the promoter regions of *gliZ*, *gliT*, *gliF* and *gtmA* in GT-inducing conditions. Using the ChIP-seq data (S4 File), we looked for motifs that were unique to or enriched in the AA condition and found 4 potential bindings sites in the *gliZ* (sense strand) promoter region, 6 potential binding sites in the *gliT* (sense strand) promoter region, whereas the *gliF* (antisense strand) and *gtmA* (sense strand) promoter regions contained 4 and 2 potential binding motifs, respectively (S2C Fig). ChIP-qPCR confirmed that RglT bound to the promoter region of *gliZ*, *gliT* and *gliF* (Fig 3F),

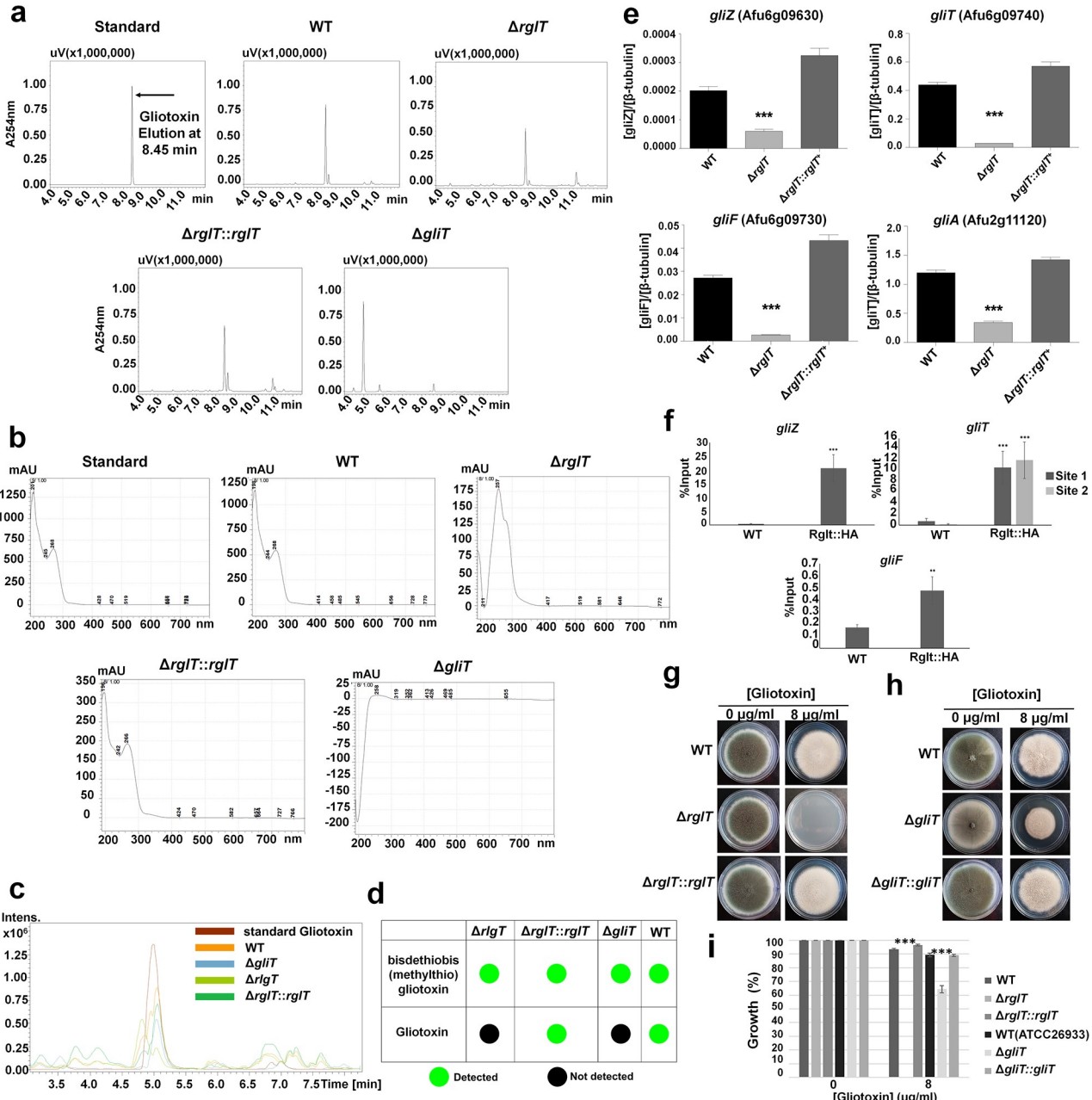

**Fig 3. RglT is crucial for gliotoxin (GT) biosynthesis. a**, HPLC (high performance liquid chromatography) analysis for GT of supernatants of different strains, that were grown for 72 h in GT-inducing conditions, showed similar chromatographs for all strains except for the Δ*gliT* strain (negative control). Standard GT (positive control) eluted after 8.45 min at 254 nm. All experiments were performed in biological triplicates. **b**, UV-vis spectra of the Δ*rglT* and Δ*gliT* strains differ from the spectra of the wild-type (WT) and Δ*rglT*::*rglT*+ strains, indicating the absence of GT. **c**, LC-MS (Liquid Chromatography-Mass Spectrometry) of biological triplicates, confirms the absence of GT from the supernatant of the Δ*rglT* strain. Shown are the extended chromatograms of organic extracts from: (1, brown) standard gliotoxin; (2, orange) WT; (3, dark green) Δ*rglT*::*rglT*; (4, pale green) Δ*rglT*; (5, blue) Δ*gliT*. **d**, The Δ*rglT* strain does not produce GT. Diagram summarizing in which strains GT or BmGT [bisdethiobis(methylthio) gliotoxin] was detected. **e**, RglT is essential for *gliZ*, *gliT*, *gliF* and *gtmA* expression in GT-inducing conditions. RT-qPCR was performed after strains were grown for 72 h in GT-inducing conditions. Gene expression was normalized by the β-tubulin-encoding gene. Standard deviations represent three biological replicates (**P-value < 0.001, ***P-value < 0.0001 in a one-way ANOVA test). **f**, RglT binds to the promoter regions of *gliZ*, *gliT* and *gliF* in GT-inducing conditions. ChIP (chromatin-immunoprecipitation)-qPCR was performed for the WT and RglT::HA strains when grown for 48 h in GT-inducing conditions. Binding motifs were identified by ChIP-sequencing and binding was calculated using the % input method, normalizing the RglT::HA-specific binding by the total cellular DNA (input). Standard deviations represent three biological replicates (**P-value < 0.001, ***P-value < 0.0001 in a paired, one-tailed student t-test). **g-i**, Endogenous GT resistance is completely abolished in the Δ*rglT* strain. Strains were grown for 5 days at 37˚C on glucose minimal medium supplemented with or without GT (g, h). Graphs (i) show the percentage of growth in the presence of GT

when compared to the control, no GT condition and standard deviations represent biological triplicates (***P-value < 0.0001 in a two-way ANOVA test).

suggesting that RglT regulates *gliZ*, *gliT* and *gliF* expression directly in conditions that result in GT production (Fig 3F).

MEME (Multiple EM for Motif Elicitation)-ChIP analysis was also carried out on the 500 bp region surrounding the peaks identified in our ChIP-seq data in order to determine conserved RglT DNA binding motifs that were specifically enriched in the presence of AA (S4 File and S2D Fig). Two related but different DNA motifs (5'-TCGG-3' and 5'-CGGNCGG-3') are specifically enriched among unique and stronger RglT:3xHA binding sites in the presence of AA, as opposed to a GC rich motifs among the peaks with similar binding under both conditions (S2D Fig). In addition, there is also a difference in dyad composition enriched between Control-enriched and AA-enriched RglT:3xHA binding classes (S2E and S2F Fig).

## RglT is crucial for protection against exogenous GT

GT MIC assays showed that the Δ*rglT* strain was sensitive to exogenously added GT, probably due to *gliT* expression being severely reduced in this strain in the presence GT-producing conditions. *gliT* encodes a GT oxidoreductase that is crucial for GT disulphide bridge formation and cleavage as well as for protection against exogenous GT[12]. To further explore the role of RglT and GliT in mediating GT self-protection, the respective deletion strains were grown on solid minimal medium supplemented with GT. The Δ*rglT* strain lost all ability to protect itself against exogenously added GT (Fig 3G and 3I). The Δ*gliT* strain was also sensitive to GT but not to the same levels as the Δ*rglT* strain (Fig 3H and 3I). These results suggest that RglT is essential for mediating protection against exogenous GT, probably through an unidentified mechanism that likely involves regulation of *gliT* and *gtmA* expression.

## The Δ*rglT* strain is hypovirulent in a chemotherapeutic murine model of IPA

Due to the importance of GT for fungal survival, colonization and immunomodulation of the mammalian host[10,11], the virulence of the Δ*rglT* strain was assessed both *in vitro* and *in vivo*. Macrophages contribute to innate immunity, fungal clearance and the generation of a pro-inflammatory response during *A*. *fumigatus* infection[23]. The capacity of bone marrow-derived macrophages (BMDMs) to phagocytize and kill wild-type (WT), Δ*rglT* and Δ*rglT*::*rglT* conidia was determined and the Δ*rglT* strain was found to be significantly more susceptible (P-value < 0.05) to macrophage-mediated phagocytosis (Fig 4A) and killing (Fig 4B).

Next, virulence of all three strains was assessed in a clinically-relevant chemotherapeutic BALB/c murine model of IPA. The WT and Δ*rglT*::*rglT* strains were both equally virulent (*p*-value > 0.05) and killed all mice after 11 and 8 days post-infection (p.i.), respectively; whereas the Δ*rglT* strain was hypovirulent [*p*-value > 0.05 when compared to the phosphate buffer saline (PBS) negative control] with 80% of mice infected with this strain being alive after 15 days p.i. (Fig 4C). Fungal burden in murine lungs showed a significant reduction of Δ*rglT* strain presence/growth in comparison with the WT and Δ*rglT*::*rglT*+ strains that was not significantly different from the negative PBS control (Fig 4D), suggesting that the Δ*rglT* strain was cleared from the lung tissue. In agreement, histopathology of murine lungs of animals infected with PBS or the Δ*rglT* strain showed the absence of inflammation and fungal hyphae (Fig 4E). In contrast, murine lungs of mice infected with the WT and the Δ*rglT*::*rglT*+ strains showed multiple foci of invasive hyphal growth, which penetrated the pulmonary epithelium

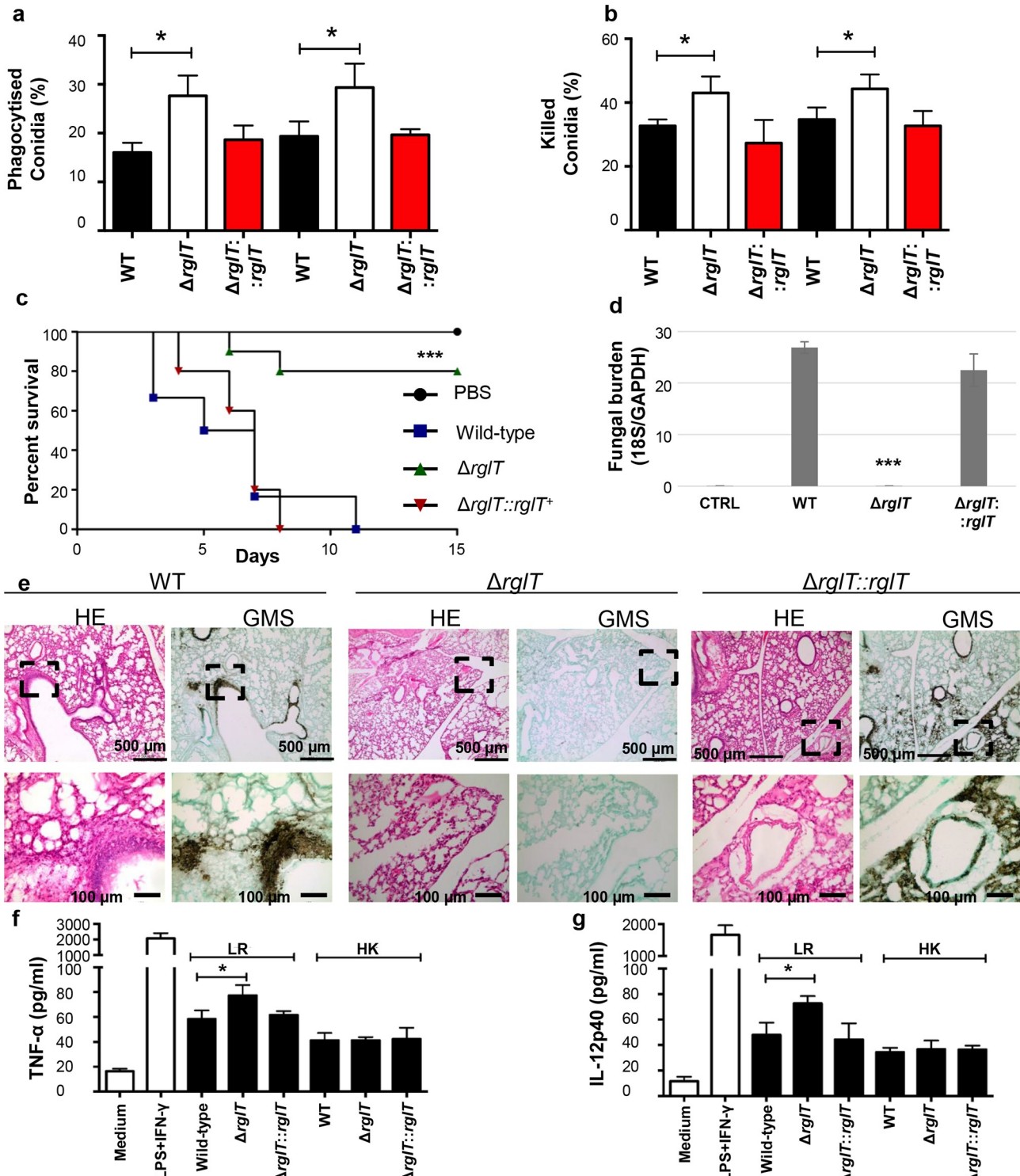

**Fig 4. RglT is essential for *A. fumigatus* virulence. a**, **b** Bone marrow-derived murine C57BL/6 (wild-type) and CGD (chronic granulomatous disease) macrophages phagocytize (**a**) and kill (**b**) a significant higher number of Δ*rglT* conidia *in vitro* than when compared to the wild-type (WT) and Δ*rglT*::*rglT* strains. Standard deviations represent three biological replicates (*P-value < 0.05 in a two-tailed, unpaired student t-test). **c**, Survival curve of chemotherapeutic BALB/c mice, which were immunosuppressed with a combination of cyclophosphamide and hydrocortisone, infected intra-nasally with the WT, Δ*rglT* and Δ*rglT*::*rglT* strains show that the Δ*rglT* strain is hypovirulent (***p-value < 0.0001 in the Mantel-Cox and Gehan-Brestow-Wilcoxon tests, comparing the Δ*rglT* to the WT and Δ*rglT*::*rglT* strains). **d,** Fungal burden of the naïve control (CTRL = phosphate buffered saline), WT, Δ*rglT* and Δ*rglT*::*rglT* strains show a significant (***p-value < 0.0001 in a two-tailed, unpaired student t-test) reduction of the Δ*rglT* growth in

chemotherapeutic murine lung tissue at 3 days post-infection. **e**, Histopathology of murine lungs at three days post-infection with the WT, Δ*rglT* and Δ*rglT*::*rglT* strains indicates the absence of Δ*rglT* proliferation in the lung tissue. 5 μm lung tissue sections were stained with haematoxylin and eosin (HE) or with Grocott's Methenamine Silver (GMS) before slides were viewed and pictures taken. **f**, Tumor necrosis factor alpha (TNF-α) and **g**, interleukin (IL)-12p40 concentrations in macrophages infected with heat killed (HK) conidia or live-resting (LR) conidia.

in the major airways, while pockets of branched invading hypha originated from the alveoli (Fig 4E). These results suggest that the absence of GT in the Δ*rglT* strain is not the major cause for the observed reduction in virulence, since GT-deficient strains, such as the Δ*gliP* strain, which is deleted for the GT nonribosomal peptide synthase, is virulent in a chemotherapeutic murine model of IPA[24].

To exclude changes in cell wall and growth defects of the Δ*rglT* strain as a possible cause for the strain-specific avirulence, macrophages were inoculated with live resting (LR) and swollen, heat-killed (HK) conidia and assayed for cytokine production. No difference in the production of TNF-α and IL-12p40 was observed between the WT, deletion and complemented strains in the presence of HK conidia; whereas LR conidia of the Δ*rglT* strain elicited a significantly greater production of both cytokines than when compared to the WT and Δ*rglT* complemented strains (Fig 4F and 4G). These results suggest that the increased macrophage proinflammatory response to the Δ*rglT* strain is not due to changes in strain-specific cell wall structure. In addition, growth of the Δ*rglT* strain on agar plates supplemented with ground porcine lung fragments and whole murine lung explants did not significantly differ from growth of the WT and Δ*rglT*::*rglT*⁺ strains (S3C and S3D Fig), suggesting that the Δ*rglT* strain grows normally on mammalian lung tissue *in vitro*.

The RNA-seq data showed that RglT regulates many genes involved in oxidation-reduction processes (S2 File), which are important for resistance against oxidative stress that is exerted by the mammalian immune system in order to fight fungal infections [25]. To determine whether oxidative stress resistance genes are under the transcriptional control of RglT, we assessed the expression of the TF *yap1* (Afu6g09930, major factor in the control of the fungal response to oxidative stress [26]), the mycelial catalase *cat1* (Afu3g02270), the superoxide dismutases *sod1* (Afu5g0924) and *sod2* (Afu4g11580), the cytosolic peroxiredoxin *prx1* (Afu4g08580), the mitochondrial peroxiredoxin *prxB* (Afu5g15070) and the thioredoxin reductase *trr1* (Afu4g12990) by qRT-PCR in the WT and Δ*rglT* strains when grown in the presence of 10 and 30 mM AA for 1 h (short incubation) and 3 h (prolonged incubation) (S4A Fig). In these conditions, RglT positively regulates the expression of *yap1*, *prxB* and *trr1*; and negatively regulates the expression of *cat1* and *sod1*, whereas no induction occurred for *sod2* and *prx1* (S4A Fig). All these genes were not significantly differently expressed after 30 min in the presence of 10 mM AA between the WT and Δ*rglT* strains (S4B Fig), suggesting that prolonged incubation times and higher concentrations of oxidative stress-inducing compounds result in a different transcriptional profile. It remains to be determined whether this regulation is direct or indirect, as the ChIP-seq data showed direct regulation of only *trr1* after 30 min incubation with 10 mM AA (S4B Fig). In addition, we grew the WT, Δ*gliP* and Δ*gliP*::*gliP* strains in the presence of oxidative stress-inducing compounds in order to further determine whether the virulence defect of the Δ*rglT* strain is due to defects in oxidative stress resistance. A relationship between gliotoxin and oxidative stress has previously been noted [11,27,28], although the mechanism has not been described to date. Growth of the Δ*gliP* strain was not significantly different from the WT and Δ*gliP* complemented strains in these conditions (S5 Fig).

To definitely determine whether the Δ*rglT* strain has defects in oxidative stress resistance, phagocytosis and killing by macrophages derived from a murine model of CGD (chronic

granulomatous disease) with NADPH deficiency was performed. CGD macrophages are still able to phagocytize (Fig 4A) and kill (Fig 4B) a significantly higher number of Δ*rglT* conidia, suggesting that the virulence defect of the Δ*rglT* strain is not due to defects in resistance to oxidative stress exerted by macrophages. Together, these results suggest that the inability of the Δ*rglT* strain to infect chemotherapeutic mice are due to unidentified mechanisms that are essential for *in vivo* survival and that have not been described *in vitro* as of date.

## The homologue of RglT has a conserved function for oxidative stress resistance and protection against exogenous GT in the non-GT-producing fungus *A. nidulans*

Secondary metabolite production is predicted to be essential for fungal survival in the highly competitive soil habitat of filamentous fungi[1], and GT is secreted by only a few *Aspergillus* species[29]. We therefore asked whether non-GT-producing *Aspergillus* species, such as *A. nidulans*, were also able to protect themselves against exogenously added GT and if so, whether this resistance against GT is mediated by RglT. Examination of the taxonomic distribution of RglT, GliT, and GT biosynthetic gene cluster (BGC) homologs across Eurotiomycetes and Sordariomycetes (Fig 5A; taxon names have been removed for clarity, but the full phylogeny is shown in S3E Fig) inferred their presence in 327, 416, and 111 taxa, respectively (S5 File). Patterns of presence or absence of the two genes harbored strong phylogenetic signal ($p < 0.001$; S5 File). Furthermore, both RglT and GliT were more commonly found in Eurotiomycetes than in Sordariomycetes. Specifically, we identified 175 / 216 (81.02%) taxa with both RglT and GliT across Eurotiomycetes, 61 of which also had a GT BGC homolog. Among Sordariomycetes, we identified 124 / 242 (51.24%) taxa with both RglT and GliT, 23 of which also had a homologous GT BGC. Lastly, phylogenetically informed model testing of the distributions of RglT and GliT showed that the pattern of occurrence of *RglT* is statistically dependent on the distribution of GliT ($p = 0.002$; S5 File) but not vice versa. This finding is consistent with an evolutionary scenario in which the GliT-based resistance mechanism is ancestral and the evolutionary recruitment of RglT regulation of GliT occurred subsequently.

According to FungiDB (https://fungidb.org/fungidb/), the orthologue of *A. fumigatus* RglT is encoded by gene AN1368 in *A. nidulans*, which has 54% identity (e-value 8e-137) at the protein level with RglT from *A. fumigatus* (Fig 5B). Similarly, FungiDB predicts *A. nidulans* gene AN3963 to encode the putative orthologue of *A. fumigatus gliT*. AN3963 has 50% identity (e-value 3e-103) at the protein level with GliT of *A. fumigatus* and is predicted to code for a putative GT reductase[12]. Indeed, *A. nidulans* grew in the presence of exogenously added GT (Fig 5C), whereas deletion of AN1368 resulted in a strain with abolished growth in the presence of GT (Fig 5C). In addition, the ΔAn1368 strain also presented reduced or abolished growth in the presence of the oxidative stress-inducing compounds AA (Fig 5D), *t*-butyl (Fig 5E), but not menadione (Fig 5F). These results suggest a conserved function of RglT for protection against oxidative stress in both *A. fumigatus* and *A. nidulans*.

To determine whether the *A. nidulans* GliT homologue-encoding gene is also transcriptionally dependent on AnRglT, qRT-PCR was performed when GT was added exogenously to the cultures. The expression of *gliT* is dependent on RglT in both *A. fumigatus* (Fig 5G) and *A. nidulans* (Fig 5H) when GT was added exogenously to the medium. In agreement with Schrettl *et al.*[12], transcriptional expression of *A. fumigatus gliT* is independent of GliZ in the presence of gliotoxin (Fig 5I). These results confirm that RglT regulates the expression of GT self-protection and resistance genes when GT is added exogenously to cultures of both *A. fumigatus* and *A. nidulans*, and suggest a conserved function of RglT for resistance against exogenous GT, even in GT non-producing *Aspergillus* species.

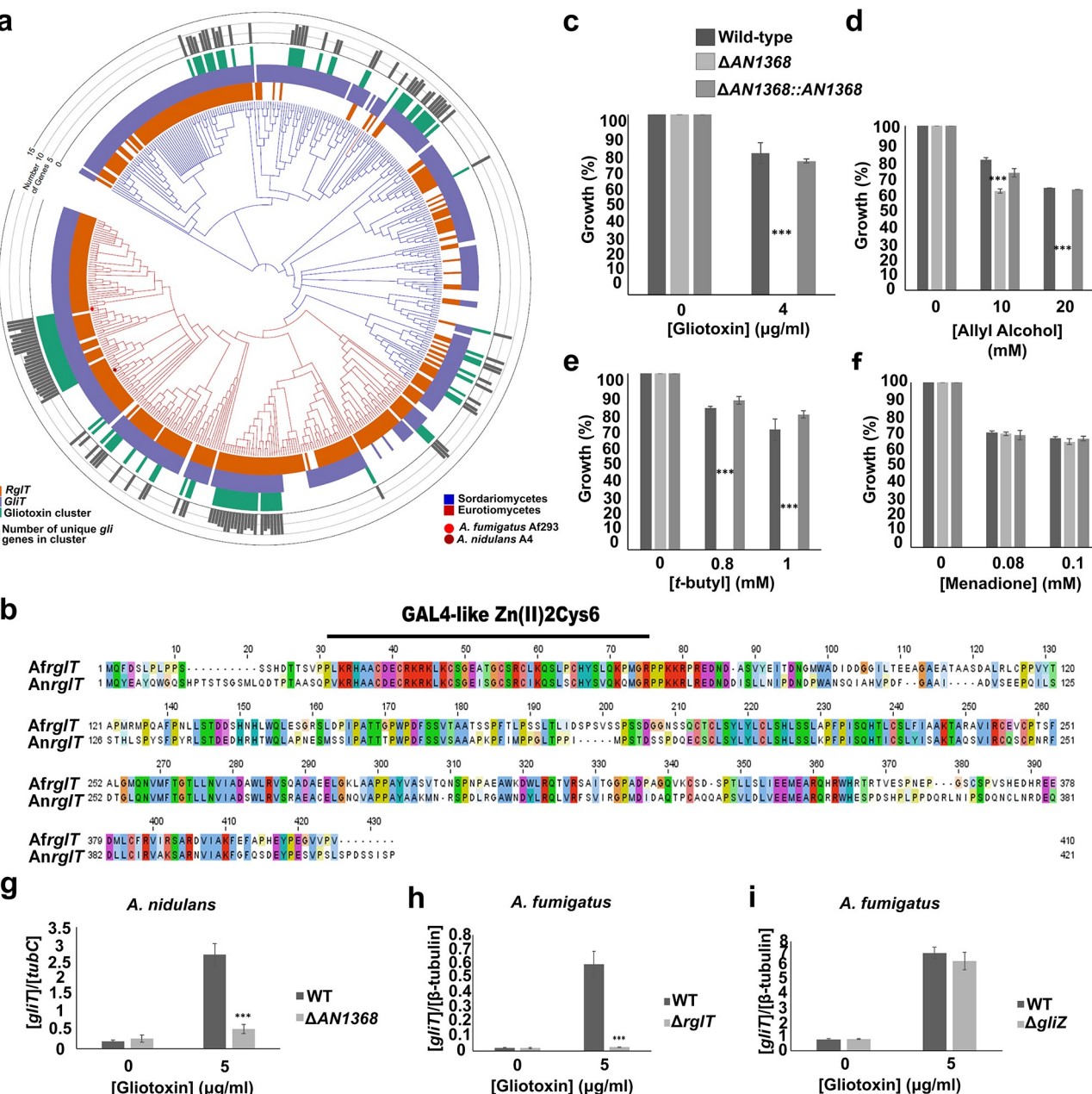

**Fig 5. The homologue of *A. fumigatus* RglT is important for gliotoxin (GT) self-protection, through regulating the expression of the *gliT* homologue in the non-GT-producing fungus *A. nidulans*. a**, The phylogenetic distribution of RglT, GliT and GT biosynthetic gene cluster homologs across 458 fungal genomes. A 3-gene phylogeny of genomes from Eurotiomycetes (shown by the red branches) and Sordariomycetes (shown by the blue branches). For every tip in the phylogeny, the presence of RglT, GliT and gliotoxin BGC homologs is depicted using orange, purple, and green bars, respectively; absences are depicted in white. The grey bar plots depict how many of the 13 *Gli* genes are present in the gliotoxin BGC homolog. The tip corresponding to the *A. fumigatus* Af293 genome is indicated by a red dot, and the tip corresponding to the *A. nidulans* A4 genome is indicated by a maroon dot. **b**, Clustal W alignment between AfRglT and AnRglT (An1368) shows 54% identity at the protein level (e-value 8e-137). **c-f**, AnRglT (AN1368) is important for resistance against oxidative stress and GT self-protection. Strains were grown for 5 days from $10^5$ spores at 37˚C on glucose minimal media supplemented with increasing concentrations of (**c**) gliotoxin, (**d**) allyl alcohol, (**e**) *t*-butyl hydro-peroxide (*t*-butyl) and (**f**) menadione. Legend for all graphs is displayed. Graphs indicate the % of growth in the presence of the respective drug with respect to the control condition (without drug). Standard deviations represent three biological replicates (*P-value < 0.01; **P-value < 0.001; ***P-value < 0.0001 in a two-way ANOVA test). **g-i**, The expression of (**g**) An*gliT* (AN3969) and (**h**) Af*gliT*, as determined by qRT-PCR, is transcriptionally dependent on RglT when GT is added exogenously for 3 h at a final concentration of 5 μg/ml. As a control, (**i**) Af*gliT* gene expression was confirmed to be independent of GliZ. Gene expression was normalised by tubulin gene expression in both fungi. Standard deviations represent three biological replicates (***P-value < 0.0001 in a two-way ANOVA test).

## Discussion

In this work we identified a TF, termed RglT (regulator of gliotoxin), that is crucial for resistance of *A. fumigatus* to oxidative stress, for gliotoxin (GT) self-protection, GT biosynthesis and virulence. We show that RglT-mediated regulation of GT biosynthesis and self-protection occurs through directly binding to the promoter region of a few *gli* genes in the presence of allyl alcohol (AA)-induced oxidative stress, as well as under GT-inducing conditions and upon the addition of exogenous GT. This includes the promoter region of *gliT*, encoding an oxidoreductase that is crucial for producing the toxic form of GT and for self-protection from GT[12]. Separate regulation of *gliT* from the other GT biosynthetic genes has been described previously[12] and this is the first study that reports on a TF that directly regulates *gliT* expression in different conditions. In agreement, deletion of *rglT* resulted in a strain that was highly sensitive to exogenously added GT and which did not produce GT, but rather the methylated form of GT, BmGT [bisdethiobis(methylthio)-gliotoxin]. BmGT formation is catalyzed by the enzyme GtmA and has been shown to play a role in the attenuation of *gli* cluster gene expression and GT biosynthesis, therefore playing a role in dithiol GT removal in *A. fumigatus*[20]. In addition to regulating GT biosynthesis, RglT is also involved in regulating the expression of *gtmA* in the presence of AA and in GT-inducing conditions. It remains to be determined to which extent RglT-mediated transcriptional regulation affects BmGT production as well as the role of RglT in the interplay of GT biosynthesis and GT production attenuation.

It is possible that RglT DNA binding is also regulated at the level of chromatin structure and/or protein-protein interaction with other DNA binding factor(s). The latter possibility is supported by the observation that two related but different DNA motifs (5'-TCGG-3' and 5'-CGGNCGG-3') are specifically enriched among unique and stronger RglT:3xHA binding sites in the presence of AA, as opposed to a GC rich motifs among the peaks with similar binding under both conditions (S2C Fig). In addition, there is also a difference in dyad composition enriched between control-enriched and AA-enriched RglT:3xHA binding classes (S2D Fig). These additional levels of regulation remain subject to future studies but would potentially result in a fine-tuning of GT biosynthesis regulation, increasing the complexity of our current understanding of GT production and attenuation. In agreement, transcriptional regulation of the GT biosynthetic gene cluster has been shown to be extremely complex, governed by a myriad of proteins[11]. Adding to this complexity, is the fact that these regulatory proteins only seem to control a subset of GT biosynthetic cluster genes[11].

Furthermore, we also observed a transcriptional down-regulation of several genes of the fumagillin biosynthetic gene cluster (BGC). The production of fumagillin is associated with the formation of another SM, pseurotin, and genes of this SM supercluster have been proposed to be physically intertwined[30]. Significant down-regulation of the fumagillin and pseurotin BGCs have been reported in the ΔgtmA strain, showing that the dysregulation of one SM BGC can have effects on the production of unrelated SM-encoding gene clusters[31]. Similarly, the deletion of *rglT* appears to not only affect gene expression of the GT BGC but also of other SM-encoding gene clusters.

The reason for *gli* gene expression in the presence of AA remains subject to further investigation but it is possible that some *gli*-encoded enzymes are important for the oxidative stress response, especially as GT biosynthesis and oxidative stress protection were shown to be somewhat intertwined in a yet to be defined mechanism[11,27]. GliT could be important for reducing other exogenous compounds, although this is unlikely as GliT was shown to be specific for GT and not dependent on other thioredoxin systems[12]. Another intriguing possibility is that the P450 monooxygenase, encoded by *gliF*, which shares a divergent promoter with *gliT*, may play a role or contribute to resistance against different oxidative stresses in *A. fumigatus*.

Cytochrome P450s (CYPs) are involved in the biosynthesis of secondary metabolites in fila-mentous fungi[32,33], although to date, GliF has not been characterized. Furthermore, a cer-tain degree of redundancy is likely to exist between these types of enzymes, with the genome of *A. fumigatus* encoding over 100 CYPs[34], thus rendering the task of identifying one specific CYP required for AA-induced oxidative stress resistance difficult.

Another significant finding is that the deletion of *rglT* results in a strain that is more sensi-tive to exogenous GT than the Δ*gliT* strain. Discrepancies of strain sensitivities to GT between this and previous work[12], may be due to different concentrations of GT that were used. Nev-ertheless, the results presented here suggest that additional mechanisms of GT resistance exist in *A. fumigatus* that remain subject to future investigations.

We also show that putative homologues of RglT are present in other fungal species and that the homologue of RglT in the non-GT-producing fungus *A. nidulans* displays functional con-servation, as a loss of An*rglT* resulted in increased sensitivity to oxidative stress and to exoge-nous levels of GT, probably due to the reduced expression of the gene encoding a putative GliT homologue in these conditions. Indeed, phylogenetically informed model testing led to an evolutionary scenario in which the GliT-based resistance mechanism is ancestral and the evolutionary recruitment of RglT regulation of GliT occurred subsequently especially in the class of Eurotiomycetes. It would therefore be of interest to investigate the functionality of the homologues of this TF and of GliT in other opportunistic mammalian pathogens as well as in fungal plant pathogens and determine a potential role in virulence, especially as the production of GT has been documented *in vivo* and shown to be crucial for fungal virulence[10],[11].

In *A. fumigatus*, RglT was shown to be essential for virulence in a chemotherapeutic murine model of invasive pulmonary aspergillosis (IPA). The observed severe reduction in virulence of the Δ*rglT* strain is not due to the absence of GT, as deletion of *gliP*, encoding the nonriboso-mal peptide synthase of the GT BGC, resulted in a strain that was still virulent in the chemo-therapeutic murine model of IPA [24]. We therefore hypothesized that the Δ*rglT* strain hypovirulence may be related to defects in oxidative stress resistance. In favor of this theory is the observation that the Δ*gliP* strain is not sensitive to the here tested oxidative stress-inducing compounds. GT is known to have immunomodulatory functions[7–9] but it also induces oxi-dative stress in mammalian cell lines and interferes with GSH levels that can exacerbate the effects of GT[11]. Enzymes encoded by the *gli* genes appear to play different roles in oxidative stress protection as observed for *gliP* and *gliK* [27]. The relationship between GT and oxidative stress is therefore extremely complex and remains subject to future investigations. The Δ*rglT* strain is sensitive to GT and oxidative stress-inducing compounds, phenotypes that can be par-tially explained by RglT controlling the expression of genes encoding enzymes, such as Yap1, required for oxidative stress resistance. Yap1 has been described as the major regulator of the *A. fumigatus* response against oxidative stress [26,35], although it is not required for virulence in a chemotherapeutic murine model of IPA [35]. This discrepancy may be explained by the fact that RglT regulates many additional genes required for oxidative stress resistance, such as a variety of oxidoreductases that have not been characterized to date. In addition, differences in regulatory mechanisms of TFs are bound to exist between the laboratory *in vitro* and murine *in vivo* conditions as has previously been shown for TFs in *A. fumigatus* [36] and *Can-dida albicans* [37]. This may also hold true for the *in vitro* macrophage assays, which showed that CGD-derived macrophages phagocytised and killed Δ*rglT* conidia with the same effi-ciency as macrophages from healthy mice, contradicting our hypothesis that oxidative stress resistance is the reason for the observed virulence deficiency of the Δ*rglT* strain. Cyclophos-phamide is less effective in impairing and depleting murine T-suppressor and T-helper cells [38], suggesting that *in vivo* additional immune cells and mechanisms may act in clearing the Δ*rglT* strain from the lung tissue. A detailed mechanism of RglT in controlling *A. fumigatus*

oxidative stress resistance and its relevance for *in vivo* virulence therefore remains to be explored in future studies. In summary, the hypovirulence of Δ*rglT* in a chemotherapeutic model of IPA is likely related to a combination of factors and results presented here warrant additional studies that are of interest for unraveling uncharacterised *A. fumigatus* virulence mechanisms.

In conclusion, this work describes the function of a previously uncharacterized transcription factor in GT biosynthesis regulation and *A. fumigatus* virulence, that may also be of importance for the virulence of other pathogenic fungal species.

## Materials and methods

### Ethics statement

Animal experiments were performed in strict accordance with the Brazilian Federal Law 11,794 establishing procedures for the scientific use of animals in strict accordance with the principles outlined by the Brazilian College of Animal Experimentation (CONCEA), and the State Law establishing the Animal Protection Code of the State of São Paulo. All protocols adopted in this study were approved by the local ethics committee for animal experiments from the Federal University of São Paulo (Permit Number: 1468200318; Studies on the interaction of *Aspergillus fumigatus* with animals). All efforts were made to minimize suffering. Animals were clinically monitored at least twice daily and humanely sacrificed if moribund (defined by lethargy, dyspnea, hypothermia and weight loss). All stressed animals were sacrificed by cervical dislocation after intraperitoneal (i.p.) injection of ketamine and xylazine.

### Strains and media

All strains used in this study are listed in S1 Table. Strains were grown at 37˚C, except for growth on allyl alcohol-containing solid medium, which was carried out at 30˚C. Complete and glucose minimal media (GMM) were prepared exactly as described previously[14]. Gliotoxin production was induced by growing the strains in Czapek-Dox (http://himedialabs.com/TD/M076.pdf) broth for 72 h. Porcine lungs were cut into small pieces and freeze-dried before they were ground under liquid nitrogen and 1 g was dissolved in a 100 ml solution of 1% w/v agarose and autoclaved. Lungs were extracted from healthy C57BL/6 mice and one lung was transferred to a 1% w/v agarose plate. All experiments were carried out in biological triplicates.

### Alcohol dehydrogenase (ADH) activity

ADH activity in total cellular protein extracts of biological triplicates of strains grown in the defined conditions was measured exactly as described previously[39].

### Microscopy

Microscopic analyses of GFP fluorescence was carried out as described previously[14], with the following modifications: strains were grown for 16 h at 30˚C before allyl alcohol was added to a final concentration of 10 mM for 15 min. Experiments were carried out in biological triplicates and nuclei of 50 hyphae were observed for GFP fluorescence for each replicate. Afu1g09190::GFP nuclear localization was calculated as a percentage in reference to the total amount of nuclei.

### Intracellular glutathione concentrations

Mycelia were ground to a fine powder in liquid nitrogen and 200 mg was re-suspended in 250 μl assay buffer (5% sulfosalicylic acid dissolved in 0.125M sodium phosphate monobasic

and 6.3 mM EDTA, pH 7.5) and centrifuged (13,000 rpm, 10 min, 4˚C). Total GSH concentrations were measured as described previously[40]. Briefly, GSH concentrations were determined with DTNB (Ellman's reagent) in the presence of glutathione reductase and NADPH. Sample glutathione concentrations were calculated from a GSH standard curve. In parallel, total cellular protein was extracted for each sample, by re-suspending 200 mg powder in 1 ml extraction buffer (50 mM Tris-HCl pH 7.0, 50 mM NaF, 1 mM DTT). Samples were centrifuged for 10 min at 13,000 rpm at 4˚C before protein concentrations were determined by Bradford assay (Bio-Rad), according to manufacturer's instructions. GSH concentrations were normalised by total intracellular protein concentrations and expressed as nmol/mg intracellular protein.

## RNA extractions, cDNA synthesis and RNA-sequencing

All experiments were carried out in biological triplicates and strains were grown from $10^6$ conidia/ml in glucose minimal medium (GMM). For the allyl alcohol (AA), strains were grown for 24 h in GMM before allyl alcohol was added to a final concentration of 10 mM for 30 min. Gliotoxin (GT) production was induced by growing strains for 72 h in Czapek-Dox medium. Alternatively, GT was added for 3 h to the culture medium to a final concentration of 5 μg/ml after strains were grown for 21 h in GMM. As GT was dissolved in DMSO, control cultures received the same volume of DMSO for 3 h. RNA was extracted as described previously[14] and cDNA was synthesized from 500 ng RNA using the ImProm-II Reverse Transcriptase (Promega), according to manufacturer's instructions. qRT-PCR was carried out on 1 μl template cDNA as described previously[14], with the exception that the total reaction volume was 10 μl (5.5 μl SYBR Green, 0.055 μl of a 100 pmol/μl primer and 3.39 μl ddH$_2$O) using the 7500 Fast Real-Time PCR Thermocycler and the 7500 Fast System v1.4.0 (AB Applied Biosystems).

For RNA-sequencing, RNA quality was verified using the Agilent Bioanalyser 2100 (Agilent technologies) using a minimum RIN (RNA Integrity Number) value of 7.0. RNA-sequencing was carried out using Illumina sequencing, as described previously[41]. The TruSeqStranded mRNA LT Set B kit (Illumina) was used for library preparation. Libraries were sequenced (2x100bp) on CTBE NGS sequencing facility HiSeq 2500 instrument, generating approx. 13.1x$10^6$ fragments per sample. Short reads were submitted to the NCBI's Sequence Read Archive under accession number SRP154617. Data processing (quality check, cleaning, removal of rRNA, genome mapping) was carried out as described previously[41], with the following modifications. Raw read counts for *A. fumigatus* genes were imported into DESeq2 using Bioconductor package tximport (version 1.12.3). DESeq2 (version 1.24.0) was used for differential gene expression quantification. Default Benjamini & Hochberg method was used for multiple hypothesis correction of DESeq2 differentially expressed genes.

## Determination of minimal inhibitory concentrations (MIC)

Strains were grown in 96-well plates at a concentration of $10^4$ spores/well in 200 μl glucose minimal medium supplemented with increasing concentrations of fumagillin or gliotoxin, according to the protocol elaborated by the Clinical and Laboratory Standards Institute (CLSI, 2017). Three independent experiments were carried out for each mycotoxin.

## Chromatin immunoprecipitation coupled to DNA sequencing (ChIP-seq)

Strains were grown in the indicated conditions and cross-linking was performed as described previously[42]. Press-dried mycelia was frozen in liquid nitrogen and stored at -80˚C until chromatin preparation, which was performed as described previously[43]. For

immunoprecipitation, 2 ug of anti-HA antibody (F7, Santa Cruz) was used. Library preparation was carried out according to[44] with the exception of the first end-repair step, which was replaced by the NEBNext End-Repair module (Cat. no. E6050). Libraries were checked and concentration determined using DNA High Sensitivity Bioanalyzer assay, mixed in equal molar ratio as described[44] and subjected to sequencing using Illumina HiSeq2500.

## ChIP-seq analysis

Raw sequencing reads were mapped to the *A. fumigatus* strain Af293 reference genome (AspGD version s03-m05-r06) using Bowtie2[45] (v2.3.5). For ChIP-seq signal data visualization purpose, aligned reads were extended to 200bp using 'macs2 pileup' command and scaled to 1 million mapped reads using 'macs2 bdgopt' command from MACS2[46] (v2.1.1) tool. UCSC Kent utils' programs[47] 'bedSort' and 'bedGraphToBigWig' were used to generate Big-Wig files, which were visualized in Integrative Genomics Viewer[48] (v2.5.3). Peaks were called using MACS2 with parameters '—nomodel—extsize 200'. MACS2 -log10(p-value) cutoff of 25 was used to select significant RglT peaks in the both AA and control ChIP-seq datasets.

Differential binding analysis was performed using DiffBind[49,50] R package (v2.12.0 with DESeq2 v 1.24.0). DiffBind combines the input peak lists into a consensus region set and performs differential binding affinity analysis over these consensus regions. Consensus regions which had peak in only one of the replicates in Control and AA condition were removed. Additionally, if a consensus region had no peak in any of the samples with -log10(p-value) > = 25, it was removed. After this quality filtering, total 538 genome-wide RglT binding sites were retained in Control and AA condition. Regions with RglT binding signal difference larger than 2-fold and FDR < = 0.05 were considered as significantly differentially bound regions between AA vs control comparison of RglT binding. Based on the DiffBind and peak occupancy analysis, RglT peaks were grouped into 5 categories such as AA specific (peak unique under AA condition), AA enriched (RglT peaks with stronger signal in AA condition compared to control), common (both conditions has RglT signal with no significant difference), control enriched (stronger signal in control compared to AA) and control specific (unique to control condition). RglT binding sites were assigned to nearest genes in *A. fumigatus* reference annotation using inhouse script.

Motif enrichment was performed on three sets of sequences, common peaks, AA enriched (AA specific + AA enriched) peaks and control enriched (control specific + control enriched) peaks. 500bp sequence around peak summit position was extracted for all significant RglT peaks and analyzed using MEME-ChIP[51] for motif enrichment. Selected motifs of interest were mapped back to the genome using FIMO[52] program from MEME-suite. Heatmap figures were made using the online program FungiExpresZ (https://cparsania.shinyapps.io/FungiExpresZ/). The ChIPseq data are available from NCBI SRA (sequence read archive) database under accession number PRJNA574873.

## ChIP-qPCR

ChIP-qPCR was carried out as described previously[14], with the exception that HA-tagged protein immunoprecipitation of 1 mg total cellular proteins was carried out as described in [53]. DNA was purified using the Qiaquick gel extraction kit (Qiagen), according to manufacturer's instructions. qPCR was run as described above.

## High-performance liquid chromatography (HPLC) and liquid chromatography-mass spectrometry (LC-MS)

All solvents used during this study were of analytical grade (Synth) or HPLC grade (Merck or J.T. Baker). The analyses of extracts were performed in a high-performance liquid chromatography

system (HPLC-Shimadzu, Kyoto, Japan), which consisted of a LC-6AD solvent pump, a SCL 10AVB system controller, a CTO-10ASVP column oven, a Rheodyne model 7725 injector, a SPD-M10AVP diode array detector (DAD), and Class VP software for data acquisition. The LC-MS analysis was performed using a UPLC (Shimadzu) coupled to Sigma-Aldrich-ascentis (2.7μm, 100mm x 4.6mm) column attached to a guard column (Agilent, 4.6 mm, 12.5 mm, 5 μm) and with the micrOTOF II mass spectrometer (Bruker Daltonics), using a gradient solvent system with acetonitrile:water with 0.1% v/v acetic acid.

All experiments were performed in biological triplicates. Extraction of 20 ml culture supernatants was performed with 3 x 50 ml chloroform and organic extracts were washed with saturated NaCl, dried with anhydrous $Na_2SO_4$ and filtered. Samples were concentrated on a rotary evaporator (Rotavapor R-300, Buchi) under reduced pressure to yield the corresponding extracts (S1 Table). Extracts (samples) were dissolved in 350 μL MeOH, while the standard gliotoxin was prepared by dissolving 200 μg gliotoxin in 350 μL MeOH and doing a 1:6 dilution (50 μL in 300 μL MeOH). Each sample was analyzed using reverse-phase HPLC (C-18 ascentis column), attached to a guard column. The gradient analysis was performed using acetonitrile:water with 0.1% v/v acetic acid. The analysis time was 35 minutes, starting from 10% to 100% acetonitrile in 30 min; then, 100% 30–32.5 min, returning to 10% 32.5–34 min, finishing in 10% 35 min; using a flow of 1ml/min. Chromatograms were analyzed at 254 nm. For LC-MS analysis, each sample was analyzed using reverse-phase HPLC (C-18 ascentis column, 2.7μm, 10cm x 4.6mm, Sigma-Aldrich). The gradient analysis was performed using acetonitrile:water with 0.1% v/v acetic acid. The analysis time was 20 minutes, starting from 30% to 70% acetonitrile in 15 min; then, 70% to 100% 15–18 min, returning to 30% 18–19 min, finishing in 30% 20 min; using a flow of 1ml/min.

## Western blotting

Total cellular protein extractions were carried out as described previously[53], and quantified using Bradford reagent (Bio-Rad), according to manufacturer's instructions. A total of 50 μg of protein per sample was run on a 4% stacking and 12% resolving, self-made gel before being transferred to a membrane as described previously[14]. Blocking, primary (1:2000 dilution of the mouse monoclonal GFP antibody, #sc-9996, Santa Cruz Biotechnology) and secondary antibody (1:2000 dilution,[14]) incubations, and membrane revelation was carried out as described previously[14].

## Bone marrow-derived macrophage (BMDM) preparation and corresponding assays

BMDMs were prepared according to [54], with modifications. Femurs and tibia from 8- to 12-week old C57BL/6 mice and gp91phox knockout (CGD—chronic granulomatous disease) mice were flushed with RPMI 1640 medium before flushed cells were cultured for 7 days in RPMI 1640 medium supplemented with 10% fetal cow serum (FCS) and 20 ng/ml M-CSF (Invitrogen). Non-adherent cells were removed, and adherent macrophages were collected and washed twice with cold PBS. Macrophage cell concentrations were determined using a Neubauer chamber. The phagocytic index was determined as described previously [55]. For fungal viability determination, 96-well plates were inoculated with 200 μl of RPMI-FCS medium containing macrophages and $1 \times 10^5$ *A. fumigatus* conidia in a 1:2 macrophage: conidia ratio. Plates were incubated at 37˚C, 5% $CO_2$ for 4 h before they were centrifuged at $400 \times g$ for 10 min at 4˚C. Supernatants were discarded and macrophage lysis was induced through the addition of $ddH_2O$. Serial dilutions of samples were carried out and plated on Sabouraud agar plates. Fungal colonies were counted after 18 h incubation at 37˚C. Cytokine

production of macrophages incubated with conidia (1:5 macrophage/conidia ratio) for 4 h at 37˚C in 5% $CO_2$ was measured as described previously [56]. Heat-killed swollen conidia were prepared by inoculating 5 x $10^6$ conidia/ml in RPMI medium for 5 h at 37˚C, before samples were centrifuged and conidia were heat-killed for 15 min at 121˚C. All experiments were carried out in biological duplicates with 4 replicates each.

## Infection of chemotherapeutic mice with *A. fumigatus*, fungal burden and histopathology

Immunosuppression of female BALB/c mice, *A. fumigatus* conidia suspension preparations, murine infections with *A. fumigatus* via intranasal instillation, fungal burden and histopathology of murine lungs were carried out exactly as described previous [56]. Briefly, mice were immunosuppressed intraperitoneally with cyclophosphamide on days -4, -1, and 2 prior to and post infection, and subcutaneously with hydrocortisonacetate on day -3 prior to infection. *A. fumigatus* conidia suspensions were prepared freshly a day prior to infection, washed three times with PBS, counted and diluted to a concentration of 5.0 x $10^6$ conidia/ ml. The viability of the administered conidia was determined by growing them in serial dilutions on YAG medium, at 37˚C. Mice (n = 10/*A. fumigatus* strain) were infected by intranasal instillation of 1.0 x $10^5$ conidia diluted in 20 μl of PBS. Mice (n = 5) which received 20 μl of PBS served as the negative control. Survival curves were carried out in two independent experiments.

To determine fungal burden in the murine lungs, mice were infected as described above, but with a higher inoculum of $1 \times 10^6$ conidia in 20 μl PBS. After 72 h, both lungs were extracted from the animals, frozen in liquid nitrogen, before DNA was extracted via the phenol-chloroform method. qPCR on the *A. fumigatus* 18S rRNA region and on the mouse GAPDH intronic region was carried out on at least 500 μg of total DNA from each sample.

For histopathology, murine lungs were fixed in 10% formaldehyde solution, diaphanized and embedded in paraffin before being sliced into 5 μm thick slices. Samples were stained with Hematoxylin and Eosin (HE) or GMS.

## Phylogenetic tree construction and alignment

To determine the taxonomic distribution of the *RglT*, *GliT*, and the gliotoxin (GT) biosynthetic gene cluster (BGC), we identified homologs of all three genetic elements across genomes in the taxonomic classes Eurotiomycetes and Sordariomycetes that we retrieved from NCBI's GenBank; only genomes with gene annotations were retrieved. From the 467 genomes that we downloaded, we removed 9 because they lacked RNA polymerase I β-subunit (*RPA2*), the largest RNA polymerase II subunit (*RNAPol2*), and the α-subunit of the translation elongation factor (*EF-1*), which we used for our fungal phylogenetic analyses (see below). This filtering resulted in the genomes of 242 Sordariomycetes and 216 Eurotiomycetes (458 total genomes).

To infer the presence or absence of *RglT*, *GliT*, and the GT BGC, we blasted the *A. fumigatus* Af293 *RglT* and *GliT* genes as well as each gene in the gliotoxin BGC against each of the 458 genomes using NCBI's Blast+ v2.3.0[57]. For all genes, we used an E-value of 1e-4 and a query coverage of 50% (*GliT* and gliotoxin BGC genes) or 40% (*RglT*). To determine the presence or absence of a homologous gliotoxin BGC, we determined the physical proximity in the genome of all homologous sequences to genes in the BGC. We considered the gliotoxin BGC to be present in a genome if 7 / 13 *Gli* genes were physically clustered and the major synthase, *GliP*, was present among the clustered genes. Physical clustering was defined using a gene distance boundary of three.

To infer the evolutionary history of the 458 taxa in Sordariomycetes and Eurotiomycetes, we employed a maximum likelihood framework to reconstruct their evolutionary history

using a concatenated matrix of *RPA2*, *RNAPol2*, and *EF*-1. Each gene was aligned using Mafft, v.7.402 [58] as previously described [59] and trimmed using trimAl, v.1.2rev59 [60], with the 'gappyout' parameter. The trimmed alignments were concatenated into a single matrix that contained 458 taxa and 3,377 amino acid sites. The phylogeny of the 458 taxa was inferred using IQ-Tree v.1.6.11 [61] with 20 independent tree searches and 10 candidate trees during each independent tree search. The best-fitting model of sequence substitution, LG+F+I+G4 [62,63], was determined automatically using Bayesian Information Criterion [64]. Bipartition support was determined using 5,000 ultrafast bootstrap approximations [65]. Poorly supported bipartitions–defined as less than 80% ultrafast bootstrap approximation support–were collapsed using TreeCollapseCL v.4 (http://emmahodcroft.com/TreeCollapseCL.html). The resulting phylogeny correctly recovered broad taxonomic relationships with the exception of *Pencillicium antarcticum* IBT 31811, which was placed among Sordariomycetes (S2F Fig). We ruled out that *P. antarcticum* IBT 31811 is a contaminant because genome-scale approaches have placed this strain among other *Penicillium* species [59]; the very long branch of this species suggests that this placement is likely a long branch attraction artifact.

To determine if *RglT*, *GliT*, and the gliotoxin were randomly distributed across the phylogeny or whether their distribution was correlated with the phylogeny, we measured the phylogenetic signal associated with the presence / absence of each genetic element using the *D* statistic [66] as implemented in the Caper, v.1.0.1 [67], package in R, v.3.5.2 (https://www.r-project.org/). To assess if there is a dependence relationship between *RglT* and *GliT* when accounting for phylogeny, we conducted model testing of four hypotheses using phytools, v.0.6–99 [68]. Specifically, we tested if (i) the pattern of occurence of *RglT* and *GliT* is independent of one another (our null hypothesis), (ii) the pattern of occurrence of *GliT* is dependent on *RglT*, (iii) the pattern of occurrence of *RglT* is dependent on *GliT*, and (iv) the patterns of occurrence of *RglT* and *GliT* are interdependent. We evaluated model fit using weighted Akaike information criterion and compared the best fitting model to the null hypothesis using a chi-squared test.

## Supporting information

**S1 Fig.** The ΔAfu1g09190 strain does not present any growth defects in non-stress conditions (**A**-,**B**) and in the presence of 2-deoxyglucose (2DG) (**C**). Strains (WT = wild-type) were grown for 5 days from $10^5$ spores at 37˚C on complete or glucose minimal medium (GMM) or on MM supplemented with xylose and increasing concentrations of 2DG, before colony diameter was measured. Standard deviations represent biological triplicates. **D**, Alcohol dehydrogenase (ADH) activity is not de-regulated in the ΔAfu1g09190 strain. Strains were grown for 24 h in fructose minimal medium (ctrl = control) before being transferred to ethanol (E)- or ethanol and glucose (GE)-rich minimal medium for different time points. Standard deviations represent biological triplicates. **E**, Afu1g09190 localises to the nucleus after the addition of allyl alcohol (AA). Three independent Afu1g09190::GFP candidates were grown for 16 h in GMM at 30˚C before 10 mM AA was added for 15 min. Germinated hyphae were viewed under a fluorescence microscope before GFP fluorescence and DAPI-stained nuclei were counted in 50 germlings and the percentage of nuclear Afu1g09190::GFP was calculated. Standard deviations represent three biological replicates (***P-value < 0.0005 in a paired, equal variance t-test comparing the AA with the control condition). **F**-**G**, The Afu1g09190 GFP- and 3xHA-tagged strains are functional and do not present growth defects. Strains were grown for 5 days from $10^5$ spores at 37˚C on GMM supplemented with increasing concentrations of AA, before colony diameter was measured. Standard deviations represent biological triplicates. **H**, The ΔAfu1g09190 strain is not impaired in intracellular glutathione (GSH) levels. Strains were grown for 24 h in glucose minimal medium (time point 0, control) before 10 mM AA was

added for different time points and GSH concentrations were determined. Standard deviations represent biological triplicates.
(TIF)

**S2 Fig. a.** Genome-wide binding signals as determined by ChIP-seq (chromatin immunoprecipitation coupled to DNA sequencing) for the wild-type (WT) and RglT(Afu1g09190)::HA strains when grown for 24 h in glucose minimal medium (control = CTRL) or after the addition of 10 mM allyl alcohol (AA) for 30 min. **b**, Gene ontology (GO) analysis of all the genes that were significantly bound by RglT::HA, as determined by ChIP-seq. Binding site preferences were divided into 3 categories: i) sites unique to or enriched for binding in the control condition, ii) sites that were bound with equal strength in both the control and AA conditions and iii) sites unique to or enriched for binding in the AA condition. Also depicted are the *p*-values for each GO category. **c**, DNA sequence and localisation of putative RglT binding motifs, based on the ChIP-seq data, that were enriched in the presence of AA, in the promoter regions of *gliT*, *gliF* and *gtmA*. Highlighted in red are the sequences that were assayed by ChIP-qPCR. **d**, MEME (Multiple EM for Motif Elicitation)-ChIP analysis of the 500 bp region surrounding the peaks identified during ChIP-seq. Shown are the sequences of potential binding motifs together with the respective *p*-value for each of the 3 categories described in (a). A trace (-) signifies that the particular binding motif was not identified in the respective condition. **e**, 3mer dyad pair enrichment analysis of 200bp summit sequences from combined peaks. GC rich 3mer dyads are significantly enriched in RglT binding region. 3mer sequences CCG| CGG and GCC|GGC are found in most of the significantly enriched dyad pairs in combined RglT peak summit sequences. Some dyad pairs had multiple variants in terms of gap sequence length between two 3mers. These gap variations for each dyad pair are shown using secondary axis in the plot. **f**, AA enriched peaks and Control enriched peaks show preference for dyad pairs with specific 3mer sequence. Dyads with CCG|CGG 3mer occur more frequently in AA specific and AA enriched RglT binding sites. Whereas dyads with GCC|GGC sequence occur more frequently in Control specific and Control enriched RglT binding sites.
(TIF)

**S3 Fig. The Δ*rglT* strain does not produce gliotoxin (GT).** HRESIMS analysis for (**a**) bisdethio-bis(methylthio)GT (BmGT) and for (**b**) GT in extracts of supernatants of strains that were grown for 72 h in GT-inducing conditions. Tables show the chemical structure, formula and mass-to-charge ratios (*m/z*) for both protonated GT and BmGT ([M+H$^+$]). **c-d**, The Δ*rglT* strain has no growth defect in the presence of porcine lung (**c**) and whole murine lung explants (**d**). The upper panel shows pictures of growth after 5 days at 37˚C from 10$^5$ spores of the wild-type, Δ*rglT* and Δ*rglT*::*rglT* strains on plates containing porcine lung or one murine lung. The lower panel depicts the graph of the radial growth from the upper panel. Standard deviations represent three biological replicates. (**e**) A three-gene phylogeny of 458 taxa in the fungal classes Eurotiomycetes and Sordariomycetes. Red branches refer to taxa from the Eurotiomycetes while blue branches refer to taxa from the Sordariomycetes. Branch lengths correspond to amino acid substitutions / site.
(TIF)

**S4 Fig. Genes encoding proteins involved in oxidative stress resistance are under the transcriptional control of RglT. (A)**, Expression of genes, as determined by qRT-PCR when the WT and Δ*rglT* strains were incubated in the presence of 10 and 30 mM allyl alcohol (AA) for 1 h and 3 h. Gene expression is given as fold induction in comparison to the control (CTRL, glucose minimal medium), AA-free condition. Gene expression was normalized by actin. Standard deviations represent three biological replicates (*P-value < 0.01; **P-value < 0.001; ***P-value < 0.0001 in a two-way ANOVA test). (B) Table summarizing gene ID, gene annotation,

log2 fold change (log2FC as determined by RNA-sequencing), and differential binding (Diffbind) FC, p-values and false discovery rates (FDRs) as determined by chromatin immunoprecipitation coupled to DNA sequencing (ChIP-seq) for the genes shown in (A).
(TIF)

**S5 Fig. Deletion of *gliP* does not increase *A. fumigatus* sensitivity to oxidative stress.** Strains were grown for 5 days from $10^5$ spores at 30˚C (allyl alcohol–AA) or 37˚C on glucose minimal media supplemented with increasing concentrations of the oxidative stress-inducing compounds (**A**) AA and (**B**) menadione (mena) and *t*-butyl hydroperoxide (*t*-butyl). Graphs indicate the % of growth in the presence of the respective drug with respect to the control condition (without drug). Graphs are the quantitation of radial growth of the pictures shown in the same panel, with standard deviations representing three biological replicates.
(TIF)

**S1 File. All significantly differently (logFC >1 or logFC < -1; adjusted P-value < 0.05) expressed genes in the wild-type (WT AA vs CTRL, tab 1) and ΔAfu1g09190 (DAfu1g09190 AA vs CTRL, tab 2) strains, comparing exposure to 10 mM allyl alcohol (AA) for 30 min with the control condition.** Also shown are the gene expression comparisons between strains in either the control (DAfu1g09190 vs WT CTRL, tab 3) or the AA (DAfu1g09190 vs WT AA, tab 4) conditions. Tabs 5 and 6 show the functional categorisation (FunCat) analyses (p-value < 0.05) of genes that were up- (tab 5) or down- (tab6) regulated in the WT strain upon the addition of AA.
(XLSX)

**S2 File. Organisation of genes that were significantly differently expressed between the wild-type (WT, CEA17) and delta Afu1g09190 in the presence of allyl alcohol (AA) into categories that are important for the response of *A. fumigatus* to oxidative stress and virulence.**
(XLSX)

**S3 File. Binding of RglT::HA to target genes that are in common (common targets) between the control and AA (allyl alcohol) conditions, unique to or enriched in the control condition (control targets) and unique to or enriched in the AA condition (AA targets).** If applicable, the significant RNA-sequencing -1 > log2FC (fold-change) > 1 values for each gene between the WT and ΔrglT strains in the control and AA conditions are also shown.
(XLSX)

**S4 File. MEME-ChIP analysis of the 500 bp region surrounding the identified peaks.**
(XLSX)

**S5 File. List of the presence or absence of RglT, GliT, and a gliotoxin (GT) BGC (biosynthetic gene cluster) among 458 Eurotiomycetes and Sordariomycetes (tab RglT GliT presence absence) and statistical summary of phylogenetic signal and phylogenetically informed RglT and GliT dependency testing (tab phylogen correlation testing).**
(XLSX)

**S1 Table. Strains used in this study.**
(DOCX)

## Acknowledgments

We thank the three reviewers and the editor for their comments and suggestions, and Drs. Robert Cramer and Thorstein Heinekamp for sending us the *A. fumigatus* Δ*gliP* and complemented strains.

## Author Contributions

**Conceptualization:** Laure N. A. Ries, Gustavo H. Goldman.

**Data curation:** Laure N. A. Ries, Lakhansing Pardeshi, Gustavo H. Goldman.

**Formal analysis:** Laure N. A. Ries, Lakhansing Pardeshi, Zhiqiang Dong, Kaeling Tan, Jacob L. Steenwyk, Jaire A. Ferreira Filho, Patricia A. de Castro, Lilian P. Silva, Nycolas W. Preite, Fausto Almeida, Renato A. C. dos Santos, Diego C. R. Hernández, Monica T. Pupo, Antonis Rokas, Flavio V. Loures, Koon H. Wong, Gustavo H. Goldman.

**Funding acquisition:** Gustavo H. Goldman.

**Investigation:** Laure N. A. Ries, Jacob L. Steenwyk, Ana Cristina Colabardini, Jaire A. Ferreira Filho, Patricia A. de Castro, Lilian P. Silva, Nycolas W. Preite, Fausto Almeida, Leandro J. de Assis, Renato A. C. dos Santos, Rebecca A. Owens, Sean Doyle, Marilene Demasi, Diego C. R. Hernández, Luís Eduardo S. Netto, Monica T. Pupo, Antonis Rokas, Flavio V. Loures, Koon H. Wong.

**Project administration:** Gustavo H. Goldman.

**Resources:** Paul Bowyer, Michael Bromley, Gustavo H. Goldman.

**Software:** Lakhansing Pardeshi.

**Supervision:** Gustavo H. Goldman.

**Writing – original draft:** Laure N. A. Ries, Gustavo H. Goldman.

**Writing – review & editing:** Laure N. A. Ries, Antonis Rokas, Koon H. Wong, Gustavo H. Goldman.

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
