## [Decision Letter · Decision Letter 0]

23 Apr 2020

Dear Professor Goldman,

Thank you very much for submitting your manuscript "The Aspergillus fumigatus transcription factor RglT is important for gliotoxin biosynthesis and self-protection, and virulence" for consideration at PLOS Pathogens. As with all papers reviewed by the journal, your manuscript was reviewed by members of the editorial board and by several independent reviewers. In light of the reviews (below this email), we would like to invite the resubmission of a significantly-revised version that takes into account the reviewers' comments.

All three reviewers and the editors are impressed with this large body of work on an unstudied transcription factor in an important human fungal pathogen. The molecular analyses of gliotoxin regulation are strong. The major finding of the study in the context of PLoS Pathogens scope is the critical role of this transcription factor in fungal virulence. However, no data into the mechanism behind the virulence defect are in the current study. A few outstanding experiments thus remain for the study to make a significant impact on our understanding of A. fumigatus pathogenesis. Reviewer 1 raises an important control and rigor question - the viability of mutant conidia in the animal model studies and the number of biological replicates of the animal experiment. Reviewer 1 makes an excellent suggestion on growing the mutant on ex planted lungs as opposed to homongenized lung tissue. Perhaps the metabolism defects of the mutant translate in vivo. Kinetic fungal burden analyses would be key. Reviewer's 2 and 3 raised the issue of oxidative stress mutants in this fungus not having virulence defects in immune compromised animal models. This is an avenue potentially worth exploring as suggested specifically by reviewer 3. Overall, the fungal molecular genetics and insights into gliotoxin regulation are strong in the study, but for PLoS Pathogens a new insight into pathogenesis is expected. We do understand that additional experiments may be delayed with COVID19 situation and if you need more time please let us know. Alternatively, one suggestion that was made is that this study might be better suited for PLoS Genetics given the strength of the molecular genetics and genomic analyses. However, if you wish to address the pathogenesis question directly, we would be happy to see this again at PLoS Pathogens. 

We cannot make any decision about publication until we have seen the revised manuscript and your response to the reviewers' comments. Your revised manuscript is also likely to be sent to reviewers for further evaluation.

Sincerely,

Robert A. Cramer

Associate Editor

PLOS Pathogens

Scott Filler

Section Editor

PLOS Pathogens

Kasturi Haldar

Editor-in-Chief

PLOS Pathogens

orcid.org/0000-0001-5065-158X

Michael Malim

Editor-in-Chief

PLOS Pathogens

orcid.org/0000-0002-7699-2064

Reviewer's Responses to Questions

**Part I - Summary**

Reviewer #1: This revised ms examines the contribution of a novel transcription factor, RglT, to the biology & virulence of AF. A deletion mutant grew normally on standard media, but was growth impaired on allyl alcohol and agents that induce oxidative stress. An extensive transcriptome and chromatin ip analysis suggested that the TF regulates genes involved in the gliotoxin biosynthesis, which was confirmed by HPLC and LC-MS. Resistance to GT was also abolished in this mutant, indicating loss of self-protection, implicating gliT. Deletion of the homolog in A. nidulans, which does not producer GT, caused similar phenotypic effect with respect to GT, allyl alcohol, and oxidative stress. A phylogenetic analysis suggested that GliT-based protection mechanisms evolved before RglT-mediated regulation of GliT.

Overall, this is an excellent body of work that delves into the function of a novel transcription factor. It is an impressive body of work that provides important information that will be of interest to the fungal community. The data is novel and the ms is clearly written.

Reviewer #2: In my review of the authors’ first submission, I felt that while the in vitro/molecular work was impressive and convincing, there were major concerns about the immunocompetent mouse model. A more conventional chemotherapy model is employed in this current submission and the result is the same i.e. the ∆rglT is attenuated for virulence. This is interesting because it has been demonstrated that gliotoxin mutants (i.e. ∆gliP) are not hypovirulent in a comparable, i.e. neutropenic, model; thus, and as the authors rightly note, the virulence phenotype ∆rglT is likely not related to its loss of gliotoxin production. While this in principle makes the ∆rglT even more intriguing, it is also a dual edge sword because there is no clear sense of why the mutant is, in fact, hypovirulent. The authors suggest that that the oxidative stress phenotype could be part of the explanation, but this remains unsubstantiated. First, and to the best of my knowledge, the link between oxidative stress sensitivity and virulence in A. fumigatus is generally not well-established. For example, the yap1 mutant, which is a global regulator of the A. fumigatus oxidative stress response (and regulated by rglT in this report), is not hypovirulent in a neutropenic model (Lessing et al., Eukaryot Cell). Second, it has already been demonstrated that the loss of gliotoxin production in A. fumigatus leads to a hypersensitivity to oxidative stress (gliK mutant). This may have not been tested for the gliP mutant of A. fumigatus, but a gliP mutant of Trichoderma virens is indeed hypersensitive to ROS (Vargas et al., 2014, Microbiology). So, we are left with the likely scenario in which the A. fumigatus gliP mutant has an oxidative stress phenotype with no virulence phenotype, at least in the this infection model.

All of the above is to say that while the manuscript provides rigorous and novel insights into gliotoxin regulation in A. fumigatus, it does not seem to provide novel insight into A. fumigatus pathogenesis. It may be difficult to elucidate the mechanism of the ∆rglT virulence defect since, as the authors mention, it is likely through a myriad of processes. One small thing the authors could do in this regard is pin down the relationship between other gliotoxin mutants and oxidative stress sensitivity. For example, if they were to demonstrate that the gliP mutant was not hypersensitive to ROS, this would strengthen the argument that the ∆rglT virulence phenotype is ROS-related.

Reviewer #3: In this report by Ries, at al the authors identify RgIT as a novel transcription factor that regulates the production of the secondary metabolite gliotoxin in Aspergillus fumigatus. The authors employ a comprehensive array of technical approaches including RNA-seq, CHIP-seq, LC-MS and multiple gene-deficient Af strains to support the conclusion that RgIT is an important regulator of the gliotoxin biosynthetic cluster. Their data further suggest that RgIT regulates gliotoxin self-protection and oxidative stress resistance. The authors have edited the paper based on previous feedback and now present a more focused and well supported studies. Primary concerns and suggestions still revolve around the in vivo studies.

The authors test the impact of RgIT to Af virulence in a model of chemotherapy-induced immunosuppression. The survival curve of treated mice supports the conclusion that RgIT deficient strain is avirulent in vivo. What remains unclear is the mechanism behind the attenuation. The ex vivo cultures with BM-Macrophages suggest that immune cells are more efficient at phagocytosing and killing the RgIT mutant. Since the in vivo study is done with drugs that impair immune function it is unclear that in vivo, innate cells are similarly able to uptake and kill the mutant better. The authors suggest that the mutant is more susceptible to oxidative stress, likely to be mediated by the activity of NADPH oxidase in vivo but there is no experimental approach done to test this possibility

**Part II – Major Issues: Key Experiments Required for Acceptance**

Reviewer #1: The 80% survival of the mice in this study, and very low fungal burden, is very impressive given the normal growth on minimal lab medium and ground porcine lung extracts (line 629). This is interpreted as the inability of the mutant to proliferate in lung tissue (line 434). Ground lung is different from lung tissue because it contains reduced substrates due to degradation of polymers. Can this strain grow effectively on a polymeric substrate like collagen or a whole lung explants in vitro? This would help separate any effects of residual host immunity in the lung from a failure to use complex substrates.

Some mutant conidia have increased loss of viability, either when they are harvested or after storage in the fridge. I may have missed this in the ms, but given the strong impact on virulence/burden, it is crucial to plate the conidial stocks used for inoculation to demonstrate no loss in viability. This is likely to have been done, but it should be stated.

I did not see how many times the strong virulence defect was repeated. There should be at least one biological replicate.

Reviewer #2: No major experiments are noted.

Reviewer #3: 1) The data shown in Fig 4D and 4E appear to indicate rapid clearance of the mutant. By day 3 there seems to be undetectable amounts of fungal burden by PCR (Fig 4D) and barely any visible conidia (4E). Including a naïve control for reference to baseline would make the data on these panels more clear by facilitating the distinction between low burden and clearance. It would also be informative to perform a kinetic analysis starting at early time points before day 3.

2) The authors employ an ex vivo culture with ground porcine lung material to argue that the mutant does not have growth defects in the presence of lung material. The experimental approach does not directly rule out in vivo growth defects in the murine lung.

3) The authors suggest that the lower virulence of the RgIT mutant might be linked to enhanced sensitivity to oxidative stress. Have the investigators examined whether the RgIT mutant can grow better in BM-macrophages from NADPH-deficient mice?

**Part III – Minor Issues: Editorial and Data Presentation Modifications**

Reviewer #1: None

Reviewer #2: 1) The authors continually refer to the rglT mutant as “avirulent”, but this seems to be inaccurate since some animals in the infection group do die. The terms “hypovirulent” or “severely attenuated for virulence” would be more appropriate.

2) The authors demonstrate that the rglT mutant conidia are more readily phagocytosed and killed by macrophages, and this corresponds with an increased pro-inflammatory (cytokine) response by macrophages. They further conclude that this phenotype is not due to cell wall abnormalities in the mutant because swollen, heat-killed conidia are not hyperinflammatory (Figure 4C). I wonder, could the mutant be germinating faster, i.e. exposing beta-glucan faster, thus leading to increased uptake and killing sensitivity relative tot he wild-type? The colonial growth of the mutant seems normal enough, but perhaps there is a subtle difference in germination kinetics. In that regard, the authors seem not to mention the incubation time ahead of cytokine analysis. Was this also 4 hrs? In any case, this could be more clearly stated. Sorry, if I’m missing something.

3) Reference 7 in the Bibliography is still missing the full title!

Reviewer #3: In lines 134-135 the authors state: “Furthermore, GT inhibits the production of pro-inflammatory cytokines, such as NFkB, by macrophages[3]” . NFkB is not a cytokine, please revise.

Throughout the paper the authors refer to the murine model as a “chemotherapeutic murine model of invasive aspergillosis” without giving any details in methods or figure legend exactly what drug treatment did they use. Please add that a combination of cyclophosphamide and hydrocortisone was used. While citing previous work for experimental detail is certainly appropriate, each paper needs sufficient detail for the reader to understand the experimental approaches used without having to read another paper.

In Figure 4B the axis is labeled as %Lethality, it would be more appropriate to label as %conidia killed.

PLOS authors have the option to publish the peer review history of their article (what does this mean?). If published, this will include your full peer review and any attached files.

Reviewer #1: No

Reviewer #2: No

Reviewer #3: No
---

## [Editor Report · Decision Letter 1]

19 May 2020

Dear Professor Goldman,

We are pleased to inform you that your manuscript 'The Aspergillus fumigatus transcription factor RglT is important for gliotoxin biosynthesis and self-protection, and virulence' has been provisionally accepted for publication in PLOS Pathogens.

One minor request of the editors is that you emphasize the virulence defect in the abstract more, it is the major and most interesting finding of the story, yet the abstract focuses too much on gliotoxin, likely tangential to the virulence phenotype.

Best regards,

Robert A. Cramer, PhD

Associate Editor

PLOS Pathogens

Scott Filler

Section Editor

PLOS Pathogens

Kasturi Haldar

Editor-in-Chief

PLOS Pathogens

orcid.org/0000-0001-5065-158X

Michael Malim

Editor-in-Chief

PLOS Pathogens

orcid.org/0000-0002-7699-2064
---

## [Editor Report · Acceptance letter]

8 Jul 2020

Dear Professor Goldman,

We are delighted to inform you that your manuscript, "The Aspergillus fumigatus transcription factor RglT is important for gliotoxin biosynthesis and self-protection, and virulence," has been formally accepted for publication in PLOS Pathogens.

Best regards,

Kasturi Haldar

Editor-in-Chief

PLOS Pathogens

orcid.org/0000-0001-5065-158X

Michael Malim

Editor-in-Chief

PLOS Pathogens

orcid.org/0000-0002-7699-2064